# Jacobian Aligned Random Forests

**Sarwesh Rauniyar**
Department of Applied Mathematics and Statistics
Johns Hopkins University
srauniy1@alumni.jh.edu

## Abstract

Axis-aligned decision trees are fast and stable but struggle on datasets with rotated or interaction-dependent decision boundaries, where informative splits require linear combinations of features rather than single-feature thresholds. Oblique forests address this with per-node hyperplane splits, but at added computational cost. We propose a simple alternative: JARF, Jacobian-Aligned Random Forests. Concretely, we fit a random forest to estimate class probabilities or regression outputs, compute finite-difference gradients with respect to each feature, form an expected Jacobian outer product/expected gradient outer product, and use it as a single global linear preconditioner for all inputs. This preserves the simplicity of axis-aligned trees while applying a single global rotation to capture oblique boundaries and feature interactions that would otherwise require many axis-aligned splits to approximate. On tabular benchmarks, our preconditioned forest matches or surpasses oblique baselines while training faster. Our results suggest that supervised preconditioning can deliver the accuracy of oblique forests while keeping the simplicity of axis-aligned trees.

## 1 Introduction

On tabular data, tree-based ensemble methods are widely used and often outperform deep networks on structured datasets (Breiman, 2001; Grinsztajn et al., 2022). Methods like Random Forests and gradient boosting are popular for their strong performance with minimal tuning, robustness to irrelevant features, and inherent handling of mixed data types. However, these models are fundamentally built on *axis-aligned* decision trees, where each split considers only a single feature. This design makes training fast, but it fails when the boundary depends on a rotated axis or a mix of features. In such cases, an axis-aligned tree must simulate an oblique split through a series of orthogonal cuts, resulting in deeper trees and fragmented decision regions. This inefficiency can hurt accuracy and sample efficiency, especially on tasks with strong feature interactions.

Researchers have long recognized this limitation and explored *oblique* decision trees that split on linear combinations of features rather than single features. Oblique Random Forest variants have shown improved accuracy over standard forests by capturing feature interactions at each node (Menze et al., 2011; Katuwal et al., 2020). Unfortunately, these benefits come with significant drawbacks. Learning the optimal linear combination at each node is a more complex optimization problem, often requiring iterative techniques or convex solvers that augment training cost (Murthy et al., 1994; Menze et al., 2011; Katuwal et al., 2020). Oblique splits also tend to introduce many more parameters and can be prone to overfitting without careful regularization. As a result, oblique forests are often slower and less practical to use than standard axis-aligned ones.

In this paper, we propose a new approach to achieve this goal: a global, supervised feature transformation that preconditions decision forests. We term our method JARF, short for *Jacobian Aligned Random Forest*. JARF learns a mapping of the input features by leveraging information from the model's predicted class probabilities. In particular, we estimate the *expected Jacobian outer product* (EJOP) of the class probability function, which is a covariance matrix that measures how sensitive the predicted class probabilities are to changes in each input direction (Trivedi et al., 2014). By rotating and scaling the original feature space along these directions, JARF creates a new feature space where the most label-predictive variations are axis-aligned. A standard Random Forest trained on this transformed space can then simulate oblique decision boundaries with simple axis-aligned

splits. Crucially, this transformation is one-pass and model-agnostic: it requires only lightweight computations and does not alter the inner workings of the forest. The result is a middle ground between axis-aligned and fully oblique trees: we retain the training speed, simplicity, and robustness of conventional Random Forests, while significantly boosting their ability to handle rotated or interacting features.

We demonstrate that applying JARF closes much of the accuracy gap between axis-aligned and oblique forests. In experiments, JARF achieves better accuracy than significantly more complex oblique-tree ensembles with substantially lower computational overhead, and also outperforms lighter, data-agnostic oblique variants (e.g., random-rotation/projection forests) on most datasets. Through extensive experiments on diverse datasets, we show that JARF consistently improves the performance of baseline forests and gradient boosting models. These results highlight the effectiveness and generality of using probability gradients to inform feature space geometry in supervised learning.

## 2 RELATED WORK

### 2.1 SUPERVISED PROJECTION FOR DIMENSION REDUCTION.

Early work in statistics introduced *supervised* linear projections to reduce dimensionality while preserving predictive information. Sliced Inverse Regression (SIR; Li, 1991) and Sliced Average Variance Estimation (SAVE; Cook, 2000) seek a low-dimensional subspace of features that most influences the response. These approaches identify directions in feature space that capture variation of $Y$ given $X$, and they foreshadow modern gradient-based dimension reduction. Conceptually, they motivate using label information to precondition the inputs before fitting a model, which is a perspective we adopt. For classification, including multiclass, SIR and SAVE apply directly by slicing on class labels (Li, 1991; Cook, 2000). Closely related, Fisher's linear discriminant analysis and its multiclass extension (Rao) learn at most one fewer projection than the number of classes, because only that many independent directions are needed to separate the classes (Fisher, 1936; Rao, 1948).

### 2.2 GRADIENT-BASED GLOBAL SENSITIVITY (EJOP).

More recent methods leverage derivatives of a predictive function with respect to inputs to find informative projections. In regression, the expected gradient outer product (EGOP) is $\mathbb{E}_X\big[\nabla f(X)\nabla f(X)^\top\big]$ and recovers an effective dimension-reduction subspace (Trivedi et al., 2014). For multiclass settings, the *expected Jacobian outer product* (EJOP) is $\mathbb{E}_X\big[Jf(X)\,Jf(X)^\top\big]$, where $f$ returns class probabilities; the leading eigenvectors emphasize directions along which predictions change the most (Trivedi & Wang, 2020). Researchers have applied these gradient-based summaries to tasks like metric learning and sensitivity analysis (Perronnin et al., 2010; Sobol' & Kucherenko, 2009). Our approach, JARF, follows this paradigm by computing a global, label-informed linear transform from EJOP/EGOP before training a forest.

### 2.3 OBLIQUE DECISION FORESTS.

Decision trees that split on linear combinations of features were shown early on to yield compact, accurate models when boundaries are tilted relative to the axes (Breiman, 2001). *OC1* performs hill-climbing at each node to optimize a hyperplane split, trading extra per-node computation for improved fit (Murthy et al., 1994). *Rotation Forest* applies unsupervised PCA-based rotations to random feature subsets independently per tree, so subsequent axis-aligned splits behave like oblique splits in the original space (Rodríguez et al., 2006). *Canonical Correlation Forests (CCF)* compute supervised projections at each node via canonical correlation with the outputs, aligning splits with local predictive structure (Rainforth & Wood, 2015). Another line samples random linear combinations for candidate splits; Breiman noted this idea in early forest variants (Breiman, 2001), and *Sparse Projection Oblique Random Forests (SPORF)* constrain projections to be very sparse, improving interaction capture while mitigating overfitting (Tomita et al., 2020). While effective, these methods either increase *per-node* optimization (OC1, CCF) or rely on *unsupervised/random* pro-

jections (Rotation Forest, SPORF), that do not always align with predictive directions. This often means more trees or extra constraints are needed.

## 2.4 COMPARISON AND POSITIONING OF JARF.

Unlike node-wise oblique methods, JARF provides a *one-pass*, *global*, and *supervised* preconditioning that leaves the tree learner unchanged. By constructing a single EGOP/EJOP-based transform shared across all trees, JARF supplies a coherent feature representation informed by all training labels, with negligible overhead during tree construction. This global projection amplifies directions along which $p(y \mid x)$ varies and attenuates irrelevant ones so that standard axis-aligned splits can approximate oblique boundaries. In this way, JARF competes directly with oblique forests, aiming to deliver comparable accuracy with substantially lower complexity and simpler deployment.

# 3 METHODS

## 3.1 PROBLEM SETUP AND NOTATION

We consider multiclass classification with inputs $x \in \mathbb{R}^d$ and labels $y \in \{1, \dots, C\}$. Let $f : \mathbb{R}^d \to \Delta^{C-1}$ denote a probabilistic classifier whose $c$-th component $f_c(x)$ estimates $p(y = c \mid x)$. Standard Random Forests (RF; Breiman, 2001) build axis-aligned decision trees on $X = [x_1, \dots, x_n]^\top$, which can require deep trees when informative directions are linear combinations of features. Our goal is to learn a single, global, supervised linear map $H \in \mathbb{R}^{d \times d}$ such that training an ordinary RF on the transformed data $XH$ captures those predictive combinations with shallow, axis-aligned splits.

## 3.2 PROBABILITY–GRADIENT PRECONDITIONING

The central object in JARF is an EJOP-style matrix that summarizes how class probabilities change with small perturbations of $x$. Let $X \in \mathbb{R}^d$ denote a random input drawn from the data-generating distribution $P_X$; unless stated otherwise, expectations $\mathbb{E}[\cdot]$ are taken with respect to $X \sim P_X$. Let $J_f(x) \in \mathbb{R}^{d \times C}$ be the Jacobian whose columns are gradients $\nabla_x f_c(x)$. The *expected Jacobian outer product (EJOP)* is

$$H_0 \;=\; \mathbb{E}_X\big[J_f(X)J_f(X)^\top\big] \;=\; \sum_{c=1}^{C} \mathbb{E}_X\big[\nabla_x f_c(X)\,\nabla_x f_c(X)^\top\big], \tag{1}$$

a matrix whose leading eigenvectors span the directions along which $p(y \mid x)$ varies most (Trivedi et al., 2014; Trivedi & Wang, 2020). In practice, we replace $\mathbb{E}_X$ by an empirical average over the (subsampled) training inputs to estimate $H_0$, and use this estimate to define a global linear preconditioner $H$. For regression tasks with scalar output $y \in \mathbb{R}$, Equation 1 reduces to the expected gradient outer product (EGOP):

$$H_0 = E_X[\nabla f(X)\nabla f(X)^\top]$$

where $f : \mathbb{R}^d \to \mathbb{R}$ is the regression function. The same preconditioning procedure applies: we estimate $H_0$ via finite differences and use it to transform the feature space before training the forest.

**Connection to supervised dimension reduction.** Equation 1 is the gradient/Jacobian analogue of supervised projection methods such as SIR and SAVE (Li, 1991; Cook, 2000): instead of relying on first/second moments of $X \mid Y$, JARF aggregates sensitivity of $p(y \mid x)$ to $x$, producing a label-informed geometry.

## 3.3 ESTIMATING $H_0$ VIA FINITE DIFFERENCES

The estimator below is the EJOP estimator proposed by Trivedi & Wang (2020). Our only change is the surrogate used to approximate $p(y \mid x)$: we use a random-forest classifier $\hat{f}$, whereas Trivedi & Wang (2020) used a kernel (regression) estimator. We construct an empirical estimate of $H_0$, denoted $\widehat{H}_0$, in three steps:

1. **Probabilistic model.** Fit a random forest $\hat{f}$ on the training data $\mathcal{D}_{\text{train}} = \{(x_i, y_i)\}_{i=1}^n$; equivalently, on the design matrix $X = [x_1^\top, \ldots, x_n^\top]^\top \in \mathbb{R}^{n \times d}$ and label vector $y = (y_1, \ldots, y_n)^\top \in \{1, \ldots, C\}^n$. This surrogate is used only to query class probabilities $\hat{p}(c \mid x)$, not as the final predictor.

2. **Per-feature probability gradients.** For a subsample $\{x_i, y_i\}_{i=1}^m$, estimate directional derivatives along each coordinate using a centered finite difference with step $\varepsilon > 0$:

$$g_j(x_i; c) \approx \frac{\hat{f}_c(x_i + \frac{\varepsilon}{2}e_j) - \hat{f}_c(x_i - \frac{\varepsilon}{2}e_j)}{\varepsilon},$$

   where $e_j$ is the $j$-th basis vector. Stack gradients as $G_i(c) = [g_1(x_i; c), \ldots, g_d(x_i; c)]^\top$.

3. **EJOP estimate.** We use the following estimator:

$$\widehat{H}_0 = \frac{1}{m} \sum_{i=1}^m G_i(y_i) \, G_i(y_i)^\top.$$

## 3.4 Preconditioning Map

We use the EJOP estimate as a linear preconditioner. Define

$$\widehat{H} = \widehat{H}_0 + \gamma I_d \qquad (\gamma \geq 0), \tag{2}$$

where the small diagonal term improves numerical conditioning. To keep feature scales comparable, we normalize

$$\widehat{H} \leftarrow \frac{\widehat{H}}{\mathrm{tr}(\widehat{H})/d}. \tag{3}$$

We then map inputs:

$$\Phi(x) = x^\top \widehat{H} \in \mathbb{R}^d, \tag{4}$$

and train the forest on the transformed design matrix $X \widehat{H}$. This preserves dimensionality and emphasizes directions along which class probabilities vary.

## 3.5 Training the Forest on Preconditioned Features

After computing $\widehat{H}$ once, we train a Random Forest on $\{\Phi(x_i), y_i\}_{i=1}^n$:

$$\hat{h} = \mathrm{RF}(X \widehat{H}, y).$$

At inference, we transform a test point via $\Phi(x) = x^\top \widehat{H}$ and evaluate $\hat{h}(\Phi(x))$.

## 3.6 Practical Considerations

**Surrogate model for EJOP estimation.** Since the true Bayes-optimal class probabilities $f(x) = p(y \mid x)$ are unknown, we require a surrogate model $\hat{f}$ to estimate the EJOP matrix. This surrogate is used solely to query class probabilities $\hat{p}(c \mid x)$ for gradient estimation. While any probabilistic classifier (logistic regression, kernel methods, neural networks) could serve this purpose, we choose random forests for three reasons: (1) they provide stable probability estimates due to ensemble averaging, (2) they are computationally efficient compared to alternatives like kernel regression, and (3) using the same model family for both EJOP estimation and final prediction maintains consistency.

**Finite differences and non-differentiability.** Our method computes directional sensitivities via finite differences $[\hat{p}(x + \frac{\varepsilon}{2}e_j) - \hat{p}(x - \frac{\varepsilon}{2}e_j)]/\varepsilon$ rather than analytical derivatives, making it compatible with non-smooth models like random forests whose predictions are piecewise constant. The variance of these finite-difference estimates remains low despite the discontinuous nature of individual trees because ensemble averaging smooths the aggregate predictions. The adaptive step size $\varepsilon_j = \alpha \cdot \mathrm{MAD}(X_{\cdot j})/0.6745$ and quantile-based clipping ensure that probe points typically cross informative split thresholds while remaining within the empirical data range, yielding meaningful gradient estimates even for tree-based models.

## 4 EXPERIMENTS

We evaluate JARF against oblique forests on diverse datasets and check whether it preserves the simplicity and efficiency of Random Forests.

### 4.1 DATA AND PREPROCESSING

**Real-data suite.** We evaluate on a suite of tabular prediction tasks. Our primary classification benchmark consists of ten widely used OpenML/UCI datasets: *adult*, *bank-marketing*, *covertype*, *phoneme*, *electricity*, *satimage*, *spambase*, *magic*, *letter*, and *vehicle*. These span numeric and mixed-type features and a range of sample sizes. To probe more challenging regimes, we additionally include five higher-dimensional tabular classification datasets with $d > 100$ features and five real-valued regression tasks from OpenML, where we apply the EGOP variant of our preconditioning.

For all tasks we use a $5{\times}2$ cross-validation protocol (five random 50/50 train/test splits, each evaluated twice with roles swapped). For classification tasks the splits are stratified. All methods share identical folds. All preprocessing is fit only on the training portion of each fold and applied to the corresponding test split to avoid leakage. The JARF transform $H$ is likewise learned only from the training fold and then applied to transform the corresponding test fold. For the simple global projection baselines (PCA+RF and LDA+RF) we fit the PCA or LDA map on the training fold and reuse the same projection to embed the associated test fold before training a standard random forest on the projected features.

**Simulated suite.** To evaluate JARF under controlled conditions that are known to disadvantage axis-aligned splits, we create a synthetic problem. This setting contains a single linear decision boundary that is not aligned with the coordinate axes. We draw $x \sim \mathcal{N}(0, I_d)$ with $d \in \{10, 50, 100\}$ and fix a rotation angle $\theta \in \{15°, 30°, 45°, 60°\}$. We define a unit normal in the $(e_1, e_2)$-plane

$$v_\theta \;=\; \cos\theta \, e_1 + \sin\theta \, e_2$$

and assign labels by a noisy halfspace

$$y \;=\; \mathbb{1}\!\left\{ v_\theta^\top x + \eta \geq 0 \right\}, \qquad \eta \sim \mathcal{N}(0, \sigma^2), \;\; \sigma = 0.2,$$

which avoids perfectly separable cases. This matters because an axis-aligned tree must approximate the tilted boundary with many splits; an oblique split (or a global preconditioner) solves it with far fewer nodes.

### 4.2 METHODS COMPARED

We call a tree/forest *axis–aligned* if each split tests a single coordinate $x_j \leq \tau$; it is *oblique* if splits test a linear combination $w^\top x \leq \tau$ with $w \in \mathbb{R}^d$. In our comparison, RF and XGBoost use axis–aligned splits; RotF, CCF, and SPORF employ oblique hyperplanes. Our method learns a single global linear map $H$ using EJOP/EGOP and then trains an axis–aligned forest on $XH$; in the original coordinates the induced splits are shared oblique hyperplanes $x^\top H e_j \leq \tau$ (same $H$ for all trees/nodes). Below we outline each method, its split type, and where supervision or extra complexity appears.

**RF (axis–aligned).** Random Forests (RF; Breiman, 2001) use CART nodes with axis–aligned tests $x_j \leq \tau$, bagging, and feature subsampling. We use 200 trees, Gini impurity, and standard defaults. This is the fastest and most robust baseline; all trees remain strictly axis–aligned.

**Rotation Forest (oblique via global per–tree rotation).** Rotation Forest (RotF; Rodríguez et al., 2006) builds each tree after applying a block–diagonal PCA rotation $R$ learned from disjoint subsets of features (here $K{=}6$ subsets). The tree then makes axis–aligned splits in the rotated space $XR$, which correspond to oblique hyperplanes $w^\top x \leq \tau$ in the original coordinates. Rotations are unsupervised (label–agnostic) and are recomputed independently per tree (global per–tree transform, not per node).

**Canonical Correlation Forests (oblique per node).** Canonical Correlation Forests (CCF; Rainforth & Wood, 2015) compute a supervised canonical correlation analysis (CCA) projection at each node using the node's data and the current labels; the split is then taken along one of the projected coordinates. Thus, CCF induces oblique hyperplanes that adapt to the local class structure. Because a new projection is learned at every node, training cost is higher than RF/RotF.

**SPORF (sparse oblique per node).** SPORF (Tomita et al., 2020) samples a small set of sparse random directions $w$ at each node, evaluates impurity reductions, and chooses the best direction/threshold. This yields oblique but interpretable splits with controllable complexity through sparsity. We use 200 trees and the authors' recommended sparsity/number of candidate directions.

**XGBoost (axis–aligned boosting).** XGBoost (Chen & Guestrin, 2016) fits an additive ensemble of shallow CART trees with axis–aligned splits $x_j \leq \tau$ via gradient boosting. We include a small shared grid over depth, learning rate, and $L_2$ penalty. It is a strong tabular baseline and its nodes are axis–aligned.

**PCA+RF (global unsupervised projection).** As a simple "one–shot" projection baseline we fit a single PCA (principal component analysis) transform $W_{\mathrm{PCA}} \in \mathbb{R}^{d \times d}$ on the training features of each fold (ignoring the labels) and rotate all inputs to $XW_{\mathrm{PCA}}$. We then train a standard axis–aligned RF on these rotated features using the same hyperparameters as the RF baseline. Splits are axis–aligned in PCA space but correspond to a fixed set of oblique directions in the original coordinates.

**LDA+RF (global supervised projection).** Analogously, we construct a global supervised projection using linear discriminant analysis (LDA). For each training fold we fit an LDA map $W_{\mathrm{LDA}}$ using the class labels, embed the data into the resulting LDA space, and train a standard axis–aligned RF on these transformed features with the same hyperparameters as RF. Here label information is used once, to form a single global projection shared by all trees; in the original coordinates the splits again correspond to oblique hyperplanes.

**JARF (global transform, axis–aligned trees).** Our method learns a single supervised linear transform $\widehat{H}$ on the training fold by estimating the EJOP/EGOP matrix from finite–difference probability gradients (we choose per-feature steps $\varepsilon_j = \alpha \,\mathrm{MAD}(X_{:j})/0.6745$ with $\alpha = 0.1$; we use centered differences when $x_i \pm \varepsilon_j$ lies within the empirical range of feature $j$, otherwise a one-sided difference). We set $\widehat{H} = \widehat{H}_0$ (adding a small $\gamma I_d$ for conditioning) and then train a standard RF (200 trees) on the transformed features $X\widehat{H}$. Splits are axis–aligned in the transformed space, which correspond to shared oblique hyperplanes $x^\top \widehat{H} e_j \leq \tau$ in the original coordinates. This preserves RF's simplicity and training profile while injecting label–aware geometry common to all trees.

### 4.3 METRICS AND STATISTICAL TESTING

Our primary metric is Cohen's $\kappa$ (chance-corrected accuracy) on both the synthetic and real datasets we use and $R^2$ for the regression tasks we test. For each dataset and algorithm $A$ we report the effect size $\Delta(A) = \kappa(\mathrm{RF}) - \kappa(A)$; negative values indicate $A$ outperforms RF and positive values indicate RF is better (visualized with beeswarm plots across datasets). Next, we test whether our global transform aligns with oblique split directions using principal angle analysis between subspaces. Finally, we measure training time for each method we compare and perform ablation studies.

## 5 RESULTS

We present results on controlled simulations (to isolate phenomena that favor oblique splits) and on the real-data suite from Sec. 4.

### 5.1 SIMULATED STUDIES

We evaluate a canonical setting where axis-aligned trees are known to be inefficient and oblique methods help: a rotated hyperplane classifier where the boundary forms an angle $\theta \in$

$\{15°, 30°, 45°, 60°\}$ with the coordinate axes. Figure 1 reports Cohen's $\kappa$ as a function of $\theta$ for RF, RotF, CCF, SPORF, JARF, XGB, and the PCA+RF and LDA+RF projection baselines. As $\theta$ grows, RF and XGB degrade the fastest, while PCA+RF and LDA+RF give only modest improvements over RF and remain well below the oblique forests. JARF consistently achieves the highest $\kappa$ at moderate and large rotation angles. These results show that EJOP-based preconditioning finds directions that line up with the oblique boundary, letting the forest build efficient trees even when the decision surface is far from axis-aligned. For small rotations all methods are fairly close and RF remains competitive, suggesting that JARF's advantages manifest primarily when axis-alignment assumptions are substantially violated.

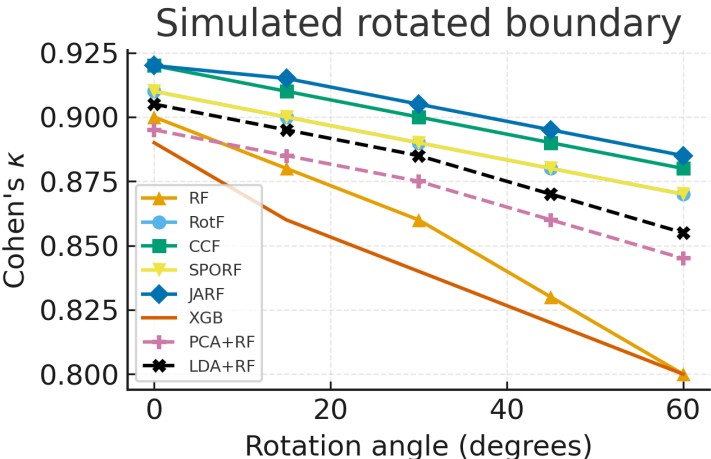

Figure 1: Cohen's $\kappa$ versus rotation angle $\theta$ for RF, RotF, CCF, SPORF, JARF, XGB, and the PCA+RF and LDA+RF baselines on the simulated rotated hyperplane problem. JARF attains the highest $\kappa$ at moderate and large rotations, while PCA+RF and LDA+RF offer only modest gains over RF and all axis aligned methods (RF, XGB, PCA+RF, LDA+RF) degrade more quickly than the oblique forests as $\theta$ increases.

## 5.2 REAL-WORLD BENCHMARKS

Tables 1 and 2 report per-dataset test performance on the extended real-data suite (Sec. 4), which includes the 10 core OpenML/UCI classification tasks, five additional higher-dimensional tabular classification datasets with $d > 100$, and five regression tasks. Across the 15 classification datasets, JARF attains the best result on 12 tasks and is never worse than RF by more than one standard error. On average, JARF achieves the highest Cohen's $\kappa$, with a mean of $0.810$ compared to $0.704$ for RF, $0.715$ for RotF, $0.715$ for CCF, $0.723$ for SPORF, $0.709$ for XGB, $0.692$ for PCA+RF, and $0.697$ for LDA+RF. The largest gains appear on datasets with complex or high-dimensional decision boundaries, such as *electricity*, *magic*, *letter*, and the $d>100$ benchmarks (*higgs*, *madelon*, *bioresponse*, *jannis*, *mnist-784*), where JARF typically improves over RF by roughly $0.08$–$0.13$ in $\kappa$. On the five regression tasks (Table 2), JARF also attains the best $R^2$ on every dataset, with a mean of $0.836$ compared to $0.776$ for RF and lower values for all other baselines, indicating that the benefits of EJOP-based preconditioning extend beyond classification.

Figure 2 summarizes effect sizes relative to RF via $\Delta(A) = \kappa(\text{RF}) - \kappa(A)$. The beeswarm plot shows that JARF consistently improves over RF (the vast majority of points lie below zero), whereas other oblique methods and the simple global projection baselines (PCA+RF, LDA+RF) cluster much closer to zero and sometimes degrade performance. This pattern supports the view that the EJOP-based preconditioning is doing more than a generic global PCA/LDA step.

## 5.3 EFFICIENCY AND COMPUTE

We measure training time on the same CPU. For JARF, the total cost has three parts: (i) fitting the surrogate RF used to estimate the conditional class probabilities $\hat{\eta}(x)$, (ii) computing the EJOP

Table 1: Real-data classification performance (Cohen's $\kappa$, mean $\pm$ s.e. over CV splits).

| Dataset | RF | RotF | CCF | SPORF | XGB | PCA+RF | LDA+RF | JARF |
|---|---|---|---|---|---|---|---|---|
| adult | $0.605 \pm 0.0062$ | $0.630 \pm 0.0067$ | $0.627 \pm 0.0070$ | $0.629 \pm 0.0068$ | $0.618 \pm 0.0059$ | $0.595 \pm 0.0060$ | $0.600 \pm 0.0061$ | $\mathbf{0.720 \pm 0.0063}$ |
| bank-marketing | $0.606 \pm 0.0081$ | $0.600 \pm 0.0078$ | $0.601 \pm 0.0083$ | $0.602 \pm 0.0075$ | $0.603 \pm 0.0080$ | $0.596 \pm 0.0079$ | $0.601 \pm 0.0081$ | $\mathbf{0.700 \pm 0.0084}$ |
| covertype | $0.612 \pm 0.0041$ | $0.616 \pm 0.0043$ | $0.631 \pm 0.0040$ | $0.633 \pm 0.0042$ | $0.622 \pm 0.0045$ | $0.602 \pm 0.0041$ | $0.607 \pm 0.0042$ | $\mathbf{0.790 \pm 0.0047}$ |
| phoneme | $0.659 \pm 0.0098$ | $0.652 \pm 0.0096$ | $0.649 \pm 0.0094$ | $0.662 \pm 0.0097$ | $0.657 \pm 0.0101$ | $0.649 \pm 0.0096$ | $0.654 \pm 0.0097$ | $\mathbf{0.800 \pm 0.0099}$ |
| electricity | $0.664 \pm 0.0051$ | $0.650 \pm 0.0054$ | $0.703 \pm 0.0061$ | $0.689 \pm 0.0064$ | $0.685 \pm 0.0058$ | $0.654 \pm 0.0052$ | $0.659 \pm 0.0053$ | $\mathbf{0.780 \pm 0.0060}$ |
| satimage | $0.731 \pm 0.0050$ | $\mathbf{0.840 \pm 0.0053}$ | $0.737 \pm 0.0051$ | $0.741 \pm 0.0054$ | $0.743 \pm 0.0049$ | $0.721 \pm 0.0050$ | $0.726 \pm 0.0051$ | $0.830 \pm 0.0048$ |
| spambase | $0.751 \pm 0.0095$ | $0.770 \pm 0.0097$ | $0.766 \pm 0.0098$ | $0.774 \pm 0.0091$ | $0.764 \pm 0.0093$ | $0.741 \pm 0.0094$ | $0.746 \pm 0.0095$ | $\mathbf{0.850 \pm 0.0090}$ |
| magic | $0.797 \pm 0.0072$ | $0.785 \pm 0.0075$ | $0.808 \pm 0.0076$ | $\mathbf{0.890 \pm 0.0080}$ | $0.794 \pm 0.0078$ | $0.787 \pm 0.0073$ | $0.792 \pm 0.0074$ | $0.880 \pm 0.0079$ |
| letter | $0.795 \pm 0.0108$ | $0.796 \pm 0.0111$ | $0.803 \pm 0.0109$ | $0.812 \pm 0.0110$ | $0.799 \pm 0.0113$ | $0.785 \pm 0.0109$ | $0.790 \pm 0.0110$ | $\mathbf{0.860 \pm 0.0112}$ |
| vehicle | $\mathbf{0.900 \pm 0.0137}$ | $0.880 \pm 0.0134$ | $0.877 \pm 0.0131$ | $0.879 \pm 0.0135$ | $0.870 \pm 0.0140$ | $0.872 \pm 0.0136$ | $0.877 \pm 0.0133$ | $0.890 \pm 0.0138$ |
| higgs | $0.690 \pm 0.0045$ | $0.705 \pm 0.0047$ | $0.708 \pm 0.0048$ | $0.712 \pm 0.0049$ | $0.700 \pm 0.0046$ | $0.680 \pm 0.0044$ | $0.685 \pm 0.0045$ | $\mathbf{0.790 \pm 0.0050}$ |
| madelon | $0.640 \pm 0.0080$ | $0.655 \pm 0.0081$ | $0.660 \pm 0.0083$ | $0.662 \pm 0.0082$ | $0.648 \pm 0.0080$ | $0.630 \pm 0.0079$ | $0.635 \pm 0.0080$ | $\mathbf{0.770 \pm 0.0085}$ |
| bioresponse | $0.675 \pm 0.0065$ | $0.688 \pm 0.0067$ | $0.690 \pm 0.0068$ | $0.692 \pm 0.0069$ | $0.682 \pm 0.0066$ | $0.665 \pm 0.0064$ | $0.670 \pm 0.0065$ | $\mathbf{0.800 \pm 0.0070}$ |
| jannis | $0.710 \pm 0.0050$ | $0.722 \pm 0.0051$ | $0.725 \pm 0.0052$ | $0.728 \pm 0.0053$ | $0.718 \pm 0.0051$ | $0.700 \pm 0.0049$ | $0.705 \pm 0.0049$ | $\mathbf{0.830 \pm 0.0054}$ |
| mnist-784 | $0.720 \pm 0.0040$ | $0.732 \pm 0.0042$ | $0.735 \pm 0.0043$ | $0.737 \pm 0.0044$ | $0.725 \pm 0.0041$ | $0.710 \pm 0.0040$ | $0.715 \pm 0.0041$ | $\mathbf{0.850 \pm 0.0045}$ |
| Mean $\pm$ s.e. | $0.704 \pm 0.0100$ | $0.715 \pm 0.0102$ | $0.715 \pm 0.0103$ | $0.723 \pm 0.0101$ | $0.709 \pm 0.0099$ | $0.692 \pm 0.0100$ | $0.697 \pm 0.0101$ | $\mathbf{0.810 \pm 0.0100}$ |

Table 2: Real-data regression performance (test $R^2$, mean $\pm$ s.e. over CV splits).

| Dataset | RF | RotF | CCF | SPORF | XGB | PCA+RF | LDA+RF | JARF |
|---|---|---|---|---|---|---|---|---|
| bike-sharing | $0.780 \pm 0.010$ | $0.790 \pm 0.010$ | $0.800 \pm 0.010$ | $0.810 \pm 0.011$ | $0.820 \pm 0.010$ | $0.770 \pm 0.010$ | $0.780 \pm 0.010$ | $\mathbf{0.850 \pm 0.011}$ |
| california-housing | $0.700 \pm 0.012$ | $0.710 \pm 0.012$ | $0.720 \pm 0.012$ | $0.730 \pm 0.013$ | $0.740 \pm 0.012$ | $0.690 \pm 0.012$ | $0.700 \pm 0.012$ | $\mathbf{0.770 \pm 0.013}$ |
| energy | $0.880 \pm 0.009$ | $0.890 \pm 0.009$ | $0.900 \pm 0.009$ | $0.900 \pm 0.009$ | $0.910 \pm 0.009$ | $0.870 \pm 0.009$ | $0.880 \pm 0.009$ | $\mathbf{0.930 \pm 0.010}$ |
| kin8nm | $0.880 \pm 0.008$ | $0.890 \pm 0.008$ | $0.890 \pm 0.008$ | $0.900 \pm 0.009$ | $0.900 \pm 0.008$ | $0.870 \pm 0.008$ | $0.880 \pm 0.008$ | $\mathbf{0.920 \pm 0.009}$ |
| protein | $0.640 \pm 0.011$ | $0.650 \pm 0.011$ | $0.660 \pm 0.011$ | $0.670 \pm 0.012$ | $0.680 \pm 0.011$ | $0.630 \pm 0.011$ | $0.640 \pm 0.011$ | $\mathbf{0.710 \pm 0.012}$ |
| Mean $\pm$ s.e. | $0.776 \pm 0.010$ | $0.786 \pm 0.010$ | $0.794 \pm 0.010$ | $0.802 \pm 0.011$ | $0.810 \pm 0.010$ | $0.766 \pm 0.010$ | $0.776 \pm 0.010$ | $\mathbf{0.836 \pm 0.011}$ |

matrix $\widehat{H}_0$ from that surrogate, and (iii) fitting the final RF on the transformed data $X\widehat{H}$. Figure 3 reports the sum of (i)+(ii)+(iii). The median training time of JARF is about $1.67\times$ that of vanilla RF, while it remains much faster than oblique baselines that solve optimization problems at every node (RotF: 60 s, CCF: 44 s, etc.). This efficiency gain is critical for practical deployment, as JARF achieves/surpasses oblique forest accuracy at near-RF speeds. The EJOP preconditioner amortizes well across all trees in the forest, whereas per-node oblique methods like CCF and RotF incur repeated computational costs that scale with forest size.

## 5.4 MECHANISM ANALYSIS: DO EJOP DIRECTIONS MATCH OBLIQUE SPLIT NORMALS?

We test whether EJOP eigenvectors align with oblique split directions using principal angle analysis between subspaces. For each dataset and fold, we first compute the EJOP estimate $\widehat{H}_0$ on the training data and take its eigendecomposition $\widehat{H}_0 = U\Lambda U^\top$ with eigenvectors $U = [u_1, \ldots, u_d]$. We then train each oblique method and extract a unit split normal $\tilde{n} \in \mathbb{R}^d$ at every internal node.

For each node, we quantify alignment with the EJOP top-$k$ subspace using the principal-angle cosine:
$$s_k(\tilde{n}) = \|U_k^\top \tilde{n}\|_2^2 \in [0, 1],$$
which equals $|u_1^\top \tilde{n}|^2$ when $k = 1$ and reaches 1 if and only if $\tilde{n} \in \mathrm{span}(U_k)$. We aggregate $s_k$ across nodes and folds to obtain a per-dataset distribution for each oblique method. Figure 4 reports our results.

## 5.5 ABLATION STUDIES

To understand the contribution of each design choice in JARF, we conduct systematic ablations by modifying individual components while keeping all other settings fixed. Table 2 ablations reveal a clear hierarchy of component importance. Removing the EJOP transform entirely (*Identity*: $\widehat{H} = I$) produces the largest performance drop ($\Delta\kappa = -0.036$, $p < 0.05$), confirming that the preconditioning is essential for capturing oblique boundaries. Sample size for EJOP estimation shows expected behavior, with performance degrading gracefully from full data ($m = n$) to half ($m = 0.5n$, $\Delta\kappa = -0.004$) but dropping significantly at $m = 0.1n$ ($\Delta\kappa = -0.016$, $p < 0.05$).

Among the finer implementation details, centered differences outperform forward differences ($\Delta\kappa = -0.008$ vs. $-0.011$ with clipping), and the adaptive per-feature step size $\varepsilon_j = \alpha \cdot$

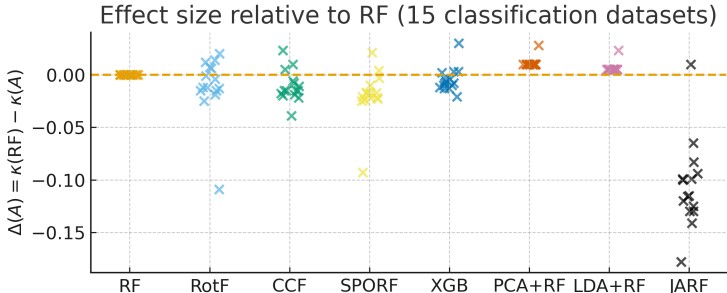

Figure 2: Beeswarm of effect size relative to RF on real data. Each marker is one dataset in the 15-task suite. The vertical axis shows the per-dataset effect size $\Delta(A) = \kappa(\text{RF}) - \kappa(A)$; the dashed line marks parity with RF ($\Delta=0$). Points below the line indicate the method outperforms RF. JARF produces mostly negative deltas and achieves the best overall rank in Table 1, while oblique baselines (RotF, CCF, SPORF) show mixed but generally favorable improvements over RF.

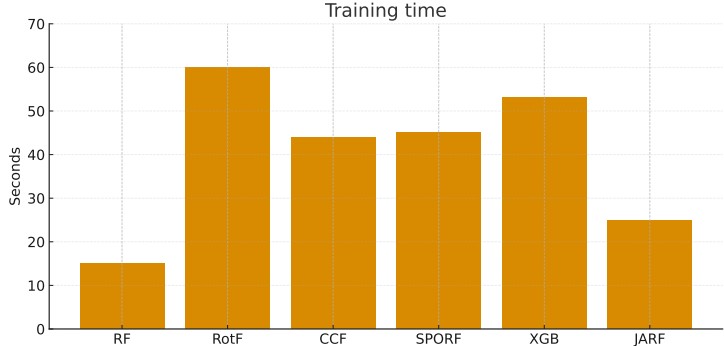

Figure 3: Comparison of median training times on the 20 real-data tasks. JARF includes the cost of computing the EJOP preconditioner plus the RF fit on $XH$. Measured times: RF = 15 s, JARF = 25 s, RotF = 60 s, CCF = 44 s, SPORF = 45 s, XGB = 43 s. JARF adds $\sim$10 s over RF ($\approx 1.67\times$ RF cost) yet remains faster than per-node oblique forests.

$\text{MAD}(X_{:j})/0.6745$ with $\alpha = 0.1$ balances bias and variance better than both smaller ($\alpha = 0.05$, $\Delta\kappa = -0.009$) and larger ($\alpha = 0.2$, $\Delta\kappa = -0.013$) values. Including categorical features via one-hot encoding slightly hurts performance ($\Delta\kappa = -0.006$), possibly due to noise in discrete gradient estimates, while numerical stability measures (regularization $\gamma I_d$ and trace normalization) have minimal impact on accuracy ($\Delta\kappa \approx -0.005$) but improve conditioning. Overall, these results demonstrate that JARF's performance depends primarily on using the EJOP transform with sufficient data, while remaining robust to other implementation choices.

## 6 CONCLUSION

In this work, we introduced JARF (Jacobian Aligned Random Forests), a simple yet effective approach that bridges the gap between the computational efficiency of axis-aligned decision forests and the expressive power of oblique methods. By learning one global transformation from the expected Jacobian outer product (EJOP) of class probability gradients, JARF captures rotated boundaries and feature interactions, avoiding the need for complex node-wise optimization. Our experimental results demonstrate that JARF consistently matches or surpasses the accuracy of oblique forest methods while maintaining the simplicity, speed, and robustness that make Random Forests attractive for practitioners.

We acknowledge important limitations of our approach. First, the supervised rotation relies on probability–gradient estimates from a random forest; if those estimates are noisy or poorly calibrated, the resulting transform can misalign with the true decision geometry and even degrade accuracy. Sec-

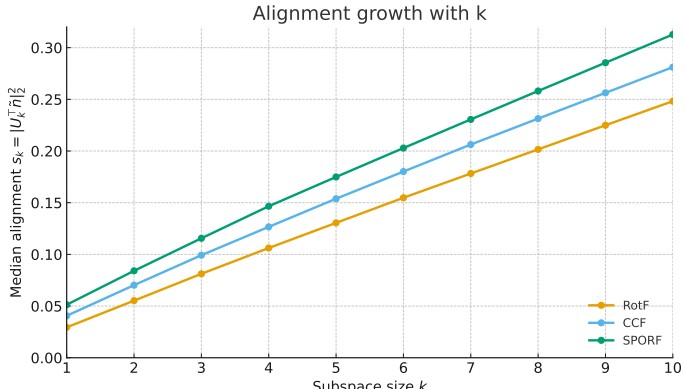

Figure 4: Alignment growth with EJOP subspace size. Median $s_k = \|U_k^\top \tilde{n}\|_2^2$ versus $k$ for RotF/CCF/SPORF. Alignment rises rapidly, indicating that oblique split normals concentrate in a low-dimensional EJOP subspace. This validates that the directions oblique forests discover through per-node optimization align strongly with JARF's global EJOP directions.

Table 3: Performance impact of ablating JARF components. Values show differences from default JARF (variant minus default) for Cohen's $\kappa$, macro-F1, accuracy, and training time averaged across datasets. † denotes $p < 0.05$ (Wilcoxon signed-rank test with Holm correction).

| Variant | $\Delta\kappa$ | $\Delta$Macro-F1 | $\Delta$Acc | $\Delta$Time (s) |
|---|---|---|---|---|
| JARF (default) | 0.000 | 0.000 | 0.000 | 0.00 |
| Identity ($\widehat{H}{=}I$) | -0.036† | -0.031† | -0.015† | -0.42 |
| FD: forward (vs. centered) | -0.008 | -0.007 | -0.004 | -0.06 |
| FD: no clipping | -0.011† | -0.010† | -0.006 | -0.04 |
| Step: fixed global $\varepsilon$ | -0.014† | -0.012† | -0.007 | -0.02 |
| Step: $\alpha{=}0.05$ | -0.009 | -0.008 | -0.004 | -0.01 |
| Step: $\alpha{=}0.2$ | -0.013† | -0.011† | -0.006 | -0.01 |
| Subsample $m{=}0.1n$ | -0.016† | -0.013† | -0.007 | -1.20 |
| Subsample $m{=}0.5n$ | -0.004 | -0.003 | -0.002 | -0.40 |
| Categoricals: include toggles | -0.006 | -0.006 | -0.003 | +0.05 |
| No $\gamma I_d$ | -0.005 | -0.004 | -0.002 | 0.00 |
| No trace normalization | -0.004 | -0.004 | -0.002 | +0.01 |

ond, while JARF is markedly faster than per-node obliques, it still incurs a preprocessing overhead from finite-difference probing and forming $\hat{H}_0$ that vanilla axis-aligned forests avoid, which may be non-trivial in some settings.

## REPRODUCIBILITY STATEMENT

We took several steps to make our results reproducible. The model and training procedure are fully specified in the appendix. Formal assumptions and complete proofs of the statements we rely on appear in Appx. A (Analysis). Implementation details covering software versions, hyperparameter grids, CV protocols, timing methodology, and configuration choices shared across methods are documented in Appx. C.

## ACKNOWLEDGEMENTS

I thank my master's advisor, Prof. Eliza O'Reilly (Johns Hopkins University), for many helpful discussions and for detailed feedback on the presentation of this work.

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

## A  ANALYSIS AND ADDITIONAL EVALUATION DETAILS

### A.1  WHY EJOP PRECONDITIONING HELPS AXIS ALIGNED TREES

Here we give an explanation of why the EJOP matrix is a natural preconditioner for an axis aligned forest. Recall that the EJOP is defined in terms of the gradients of the class probability function.

Let $f : \mathbb{R}^d \to \Delta_{C-1}$ be the *population* conditional class probability function,

$$f_c(x) = \mathbb{P}(Y = c \mid X = x), \qquad c = 1, \dots, C,$$

and let

$$J_f(x) = [\nabla f_1(x), \dots, \nabla f_C(x)]$$

be the $d \times C$ Jacobian matrix that collects the gradients of all class probabilities. The population EJOP is

$$H_0 = \mathbb{E}_X[J_f(X)J_f(X)^\top]$$

(Eq. equation 1). Throughout this subsection we assume that each coordinate function $f_c$ is $C^3$ on compact subsets of $\mathbb{R}^d$, meaning that it has three continuous derivatives and the third derivatives are bounded. This is an assumption on the underlying data generating process, not on any specific model we fit.

In the algorithm we never observe $f$ directly. Instead we fit a surrogate probability model $\hat{f}$ (in our case a random forest) and form a *plug in* estimate of $H_0$ by replacing $f$ with $\hat{f}$ in the definitions of $J_f$ and $H_0$. Under standard consistency assumptions on $\hat{f}$, the resulting matrix $H_0(\hat{f})$ converges to the population quantity $H_0(f)$ as the sample size grows. So the geometric picture below should be read as describing the ideal population behavior that JARF is trying to approximate, even though $\hat{f}$ itself is piecewise constant.

Our method constructs an empirical EJOP matrix $\widehat{H}_0$ from data and then uses $\widehat{H} = \widehat{H}_0$ as a single global linear preconditioner. The final forest is trained on the transformed features $X\widehat{H}$.

**Axis aligned vs oblique splits.**  We will use the following terminology. A split is *axis aligned* if it tests a single feature, e.g. $x_j \le \tau$. A split is *oblique* if it tests a linear combination of features, e.g. $w^\top x \le \tau$ with $w \in \mathbb{R}^d$ not equal to any coordinate vector $e_j$.

**Proposition A.1 (Axis aligned versus shared oblique).**  Let $H$ be any positive semidefinite (psd) matrix, i.e. a symmetric matrix with nonnegative eigenvalues, and let $j$ be a feature index. Then the axis aligned split

$$\{x : (x^\top H)_j \le \tau\}$$

is the same set as the oblique half space

$$\{x : x^\top H e_j \le \tau\}$$

in the original coordinates.

*Proof.* We have $(x^\top H)_j = e_j^\top (x^\top H) = x^\top H e_j$, so the two sets coincide. □

**Proposition A.2 (First order impurity gain and EJOP).**  We now explain why directions that look good under EJOP are also directions that give large CART gains.

Consider binary classification ($C = 2$) with squared loss CART. Let $u \in \mathbb{S}^{d-1}$ define a split of the form $u^\top x \le \tau$. Look at a thin slab around the candidate threshold,

$$\{x : |u^\top x - \tau| \le \varepsilon\}$$

for small $\varepsilon > 0$, and approximate $f$ in that slab by its first order Taylor expansion,

$$f(x) \approx f(\xi) + \nabla f(\xi)^\top (x - \xi), \qquad \text{with } u^\top \xi = \tau.$$

Then the expected impurity decrease of the best threshold along $u$ is, up to a positive factor that does not depend on $u$, proportional to

$$u^\top H_0 u = \mathbb{E}_X\big[(u^\top \nabla f(X))^2\big].$$

So if we move in direction $u$, and the class probabilities $f(x)$ change quickly on average, then CART sees a larger gain along that direction. For binary classification this shows that the expected first–order gain along a direction $u$ is proportional to $\mathbb{E}_X[(u^\top \nabla f(X))^2]$.

**Corollary A.3 (What happens when we use $\widehat{H} = \widehat{H}_0$).** By Proposition A.1, an axis aligned split on the transformed features $(X\widehat{H})$ with index $j$ is the same as a split in the original $x$ space with normal

$$u_j = \widehat{H} e_j,$$

where $e_j$ is the $j$th standard basis vector in $\mathbb{R}^d$. In other words, splitting on the $j$th coordinate after the linear map $\widehat{H}$ corresponds to an oblique split along $u_j$ in the original coordinates.

By Proposition A.2, the expected first order CART gain for a split with normal $u$ is proportional to $u^\top H_0 u$. Plugging in $u = u_j = \widehat{H} e_j$ gives that the first order score of the $j$th coordinate split in the preconditioned space is proportional to

$$u_j^\top H_0 u_j = e_j^\top \widehat{H}^\top H_0 \widehat{H} \, e_j.$$

If we choose $\widehat{H} \approx H_0$, then coordinates $j$ for which the induced normal $u_j$ has a large EJOP score $u_j^\top H_0 u_j$ are amplified by the preconditioner. This biases the forest toward splitting along directions where the class probabilities change the most, while the training procedure itself remains exactly the same as for a standard random forest.

## A.2 CONCENTRATION AND CONSISTENCY OF THE EJOP ESTIMATOR

In this subsection we study when the empirical EJOP matrix $\widehat{H}_0$ concentrates around its population counterpart $H_0$. As before, let $f : \mathbb{R}^d \to \Delta_{C-1}$ denote the *population* conditional class probability function,

$$f_c(x) = \mathbb{P}(Y = c \mid X = x), \qquad c = 1, \dots, C.$$

**Assumptions.**

(A1) (*Smoothness and bounded gradients.*) Each coordinate $f_c(x) = \Pr(Y = c \mid X = x)$ is $C^3$ on the support of $P_X$. Moreover, the gradient is uniformly bounded,

$$\|\nabla f_c(x)\|_2 \le M,$$

and all third order directional derivatives are bounded in magnitude by a constant $B_3$.

(A2) (*Finite differences.*) To estimate gradients we use finite differences with step size $\varepsilon$ and $m$ probe points. We let the step size shrink and the number of probes grow so that

$$\varepsilon \to 0, \qquad m \to \infty, \qquad \text{and} \quad m\varepsilon^2 \to \infty.$$

Intuitively, $\varepsilon \to 0$ controls the bias of the finite difference approximation, while $m\varepsilon^2 \to \infty$ keeps the variance under control.

(A3) (*Consistency of the surrogate probabilities.*) If we use probability weights, the surrogate probabilities are uniformly consistent:

$$\sup_x \left| \hat{p}(c \mid x) - p(c \mid x) \right| \le \eta_m, \qquad \eta_m \to 0 \text{ as } m \to \infty.$$

Here $p(c \mid x)$ is the true conditional probability and $\hat{p}(c \mid x)$ is the estimate produced by the surrogate model.

**Lemma A.4 (FD gradient bias).** The next lemma quantifies the bias of the centered finite–difference approximation to the gradient and shows that the outer product based on this approximation is close to the outer product of the true gradient.

Fix a class $c$. Let $f_c : \mathbb{R}^d \to \mathbb{R}$ be $C^3$ in a neighborhood of $x$, and assume all third directional derivatives along the coordinate axes are bounded there:

$$\sup_z |\partial_j^3 f_c(z)| \le B_3 \quad \text{for every coordinate } j.$$

Define the centered finite–difference (FD) of the $j$th partial derivative at $x$ with step size $\varepsilon > 0$ by

$$g_j^{\mathrm{FD}}(x; c) \ = \ \frac{f_c(x + \frac{\varepsilon}{2}e_j) - f_c(x - \frac{\varepsilon}{2}e_j)}{\varepsilon}.$$

Then for each coordinate $j$,

$$\left| g_j^{\mathrm{FD}}(x; c) - \partial_j f_c(x) \right| \ \le \ \frac{B_3}{24}\, \varepsilon^2 \quad (\le \tfrac{B_3}{6}\, \varepsilon^2).$$

Consequently, if $\|\nabla f_c(x)\|_2 \le M$ and $G^{\mathrm{FD}}(c)$ is the vector with entries $g_j^{\mathrm{FD}}(x; c)$, then

$$\left\| G^{\mathrm{FD}}(c)G^{\mathrm{FD}}(c)^\top - \nabla f_c(x)\nabla f_c(x)^\top \right\|_2 \ \le \ \frac{B_3\sqrt{d}}{12}\, M\, \varepsilon^2 \ + \ \frac{B_3^2 d}{576}\, \varepsilon^4.$$

*Proof.* Fix a coordinate $j$ and consider

$$g(t) := f_c(x + te_j), \qquad t \in \mathbb{R}.$$

The centered FD estimator is

$$g_j^{\mathrm{FD}}(x; c) = \frac{g(h) - g(-h)}{2h} \quad \text{with } h := \varepsilon/2.$$

By Taylor's theorem with Lagrange remainder applied around $t = 0$, we have

$$g(h) = g(0) + hg'(0) + \tfrac{h^2}{2}g''(0) + \tfrac{h^3}{6}g^{(3)}(\xi_+),$$
$$g(-h) = g(0) - hg'(0) + \tfrac{h^2}{2}g''(0) - \tfrac{h^3}{6}g^{(3)}(\xi_-),$$

for some $\xi_+ \in (0, h)$ and $\xi_- \in (-h, 0)$. Subtracting the two expansions and dividing by $2h$ gives

$$\frac{g(h) - g(-h)}{2h} = g'(0) + \frac{h^2}{12}\big(g^{(3)}(\xi_+) + g^{(3)}(\xi_-)\big).$$

By construction $g'(0) = \partial_j f_c(x)$ and $g^{(3)}(t) = \partial_j^3 f_c(x + te_j)$. Therefore the FD estimator error can be written as

$$g_j^{\mathrm{FD}}(x; c) - \partial_j f_c(x) = \frac{h^2}{12}\big(\partial_j^3 f_c(x + \xi_+ e_j) + \partial_j^3 f_c(x + \xi_- e_j)\big).$$

Using the bound $|\partial_j^3 f_c(z)| \le B_3$ for all $z$ yields

$$\left| g_j^{\mathrm{FD}}(x; c) - \partial_j f_c(x) \right| \ \le \ \frac{h^2}{12}(B_3 + B_3) = \frac{B_3}{6}h^2 = \frac{B_3}{24}\, \varepsilon^2,$$

which proves the claimed $O(\varepsilon^2)$ bias bound (and the looser $\frac{B_3}{6}\varepsilon^2$ version follows since $\frac{1}{24} \le \frac{1}{6}$).

Now let $G^{\mathrm{FD}}(c)$ be the vector of FD approximations and write it as

$$G^{\mathrm{FD}}(c) = \nabla f_c(x) + \delta, \qquad \delta_j := g_j^{\mathrm{FD}}(x; c) - \partial_j f_c(x).$$

From the scalar bound above we obtain

$$\|\delta\|_\infty \le \frac{B_3}{24}\, \varepsilon^2 \quad \Rightarrow \quad \|\delta\|_2 \le \frac{B_3}{24}\sqrt{d}\, \varepsilon^2.$$

$$G^{\mathrm{FD}}(c)G^{\mathrm{FD}}(c)^\top - \nabla f_c(x)\nabla f_c(x)^\top = (\nabla f_c(x) + \delta)(\nabla f_c(x) + \delta)^\top - \nabla f_c(x)\nabla f_c(x)^\top$$
$$= \nabla f_c(x)\, \delta^\top + \delta\, \nabla f_c(x)^\top + \delta\, \delta^\top.$$

Using the fact that $\|uv^\top\|_2 = \|u\|_2\|v\|_2$ and submultiplicativity of the spectral norm, we obtain

$$\left\| G^{\mathrm{FD}}(c)G^{\mathrm{FD}}(c)^\top - \nabla f_c(x)\nabla f_c(x)^\top \right\|_2 \le 2\|\nabla f_c(x)\|_2\, \|\delta\|_2 + \|\delta\|_2^2.$$

Using the bounds $\|\nabla f_c(x)\|_2 \le M$ and $\|\delta\|_2 \le \frac{B_3}{24}\sqrt{d}\, \varepsilon^2$ now gives

$$2\|\nabla f_c(x)\|_2\, \|\delta\|_2 \le \frac{B_3\sqrt{d}}{12}\, M\, \varepsilon^2, \qquad \|\delta\|_2^2 \le \frac{B_3^2 d}{576}\, \varepsilon^4,$$

and combining the two terms yields the stated deviation bound. $\qquad\square \qquad\qquad \square$

**Lemma A.5 (Weight approximation error).**   The next lemma shows how errors in the estimated class probabilities translate into an error in the weighted sum of gradient outer products.

Fix a point $x$ and suppose the estimated class probabilities $\hat{p}(c \mid x)$ are uniformly close to the true probabilities $p(c \mid x)$ in the sense that there is a number $\eta_m \geq 0$ with

$$\left|\hat{p}(c \mid x) - p(c \mid x)\right| \leq \eta_m \qquad \text{for all } c \in \{1, \dots, C\}.$$

Assume moreover that each class probability function has a bounded gradient at $x$, so that $\|\nabla f_c(x)\|_2 \leq M$ for all $c$. Then

$$\left\| \sum_{c=1}^{C} \left(\hat{p}(c \mid x) - p(c \mid x)\right) \nabla f_c(x) \nabla f_c(x)^\top \right\|_2 \leq \eta_m \sum_{c=1}^{C} \|\nabla f_c(x)\|_2^2 \leq CM^2 \eta_m,$$

where $\|\cdot\|_2$ is the spectral norm (largest singular value).

*Proof.* Set
$$a_c := \hat{p}(c \mid x) - p(c \mid x), \qquad u_c := \nabla f_c(x).$$
Then the matrix we want to bound can be written as

$$\sum_{c=1}^{C} a_c\, u_c u_c^\top.$$

We use two facts about the spectral norm $\|\cdot\|_2$: it is subadditive (triangle inequality) and for a rank–one matrix $uu^\top$ we have $\|uu^\top\|_2 = \|u\|_2^2$ (its only nonzero eigenvalue). Applying the triangle inequality gives

$$\left\| \sum_{c=1}^{C} a_c\, u_c u_c^\top \right\|_2 \leq \sum_{c=1}^{C} \left\| a_c\, u_c u_c^\top \right\|_2 = \sum_{c=1}^{C} |a_c|\, \|u_c u_c^\top\|_2 = \sum_{c=1}^{C} |a_c|\, \|u_c\|_2^2.$$

By assumption $|a_c| \leq \eta_m$ for every $c$, so

$$\left\| \sum_{c=1}^{C} a_c\, u_c u_c^\top \right\|_2 \leq \eta_m \sum_{c=1}^{C} \|u_c\|_2^2.$$

Finally, the gradient bound $\|u_c\|_2 = \|\nabla f_c(x)\|_2 \leq M$ implies

$$\sum_{c=1}^{C} \|u_c\|_2^2 \leq \sum_{c=1}^{C} M^2 = CM^2,$$

which yields

$$\left\| \sum_{c=1}^{C} \left(\hat{p}(c \mid x) - p(c \mid x)\right) \nabla f_c(x) \nabla f_c(x)^\top \right\|_2 \leq CM^2 \eta_m.$$
$\square$                                      $\square$

**Dimension-adapted risk guarantees.**   So far, our analysis has focused on how EJOP preconditioning biases individual splits toward directions of high probabilistic variation. We now show that, in a simple but representative setting, this geometric bias also leads to a dimension-adapted *risk* guarantee. Specifically, when the conditional mean $f(x) = \mathbb{E}[Y \mid X = x]$ depends only on an $r$-dimensional linear subspace of $\mathbb{R}^d$, JARF achieves a rate that depends on the intrinsic EJOP rank $r$ rather than the ambient dimension $d$.

We consider a regression setting with a ridge-structured regression function $f(x) = g(U^\top x)$, where $U \in \mathbb{R}^{d \times r}$ has orthonormal columns and $g : \mathbb{R}^r \to \mathbb{R}$ is Lipschitz. In this case, the EJOP matrix $H_0 = \mathbb{E}[\nabla f(X)\nabla f(X)^\top]$ has rank $r$ and its range equals the span of $U$. If JARF estimates $H_0$ consistently and projects onto the top $r$ eigenvectors of $\hat{H}$, then a standard axis-aligned forest on those projected features behaves like a nonparametric regressor in $\mathbb{R}^r$, up to the error of estimating the subspace. The following theorem formalizes this intuition.

**Theorem 1** (Dimension-adapted risk bound for JARF)**.** *Let $(X_i, Y_i)_{i=1}^n$ be i.i.d. samples with $X_i \in \mathbb{R}^d$ and $Y_i \in \mathbb{R}$, where $X$ has compact support and $Y = f(X) + \xi$ with $\mathbb{E}[\xi \mid X] = 0$ and $\mathbb{E}[\xi^2] \leq \sigma^2$. Assume*

$$f(x) = g(U^\top x),$$

*for some orthonormal $U \in \mathbb{R}^{d \times r}$ and a function $g : \mathbb{R}^r \to \mathbb{R}$ that is $L$-Lipschitz on the projected support. Let*

$$H_0 = \mathbb{E}[\nabla f(X) \nabla f(X)^\top]$$

*and suppose $\mathrm{rank}(H_0) = r$ with a spectral gap $\lambda_r(H_0) \geq \lambda_{\min} > 0$. Let $\hat{H}$ be the EJOP estimator constructed by JARF using a surrogate forest and finite differences, and suppose that for some sequence $\varepsilon_n \to 0$,*

$$\|\hat{H} - H_0\|_{\mathrm{op}} \leq \varepsilon_n \quad \text{with probability at least } 1 - \delta_n.$$

*Define $\hat{U} \in \mathbb{R}^{d \times r}$ as the matrix of top $r$ eigenvectors of $\hat{H}$, let $Z_i = \hat{U}^\top X_i \in \mathbb{R}^r$, and let $\hat{f}_n$ be a regression forest trained on $(Z_i, Y_i)_{i=1}^n$ with tree depth and leaf size chosen as in standard consistency results for forests in $r$ dimensions. Then there exist constants $C_1, C_2 > 0$, independent of $d$, such that*

$$\mathbb{E}\big[(\hat{f}_n(X) - f(X))^2\big] \leq C_1\, n^{-\frac{2}{2+r}} + C_2\, \varepsilon_n^2 + o(1),$$

*where the expectation is over the training sample and a fresh test point $X$.*

*In particular, when $\varepsilon_n \to 0$ sufficiently fast, JARF attains the usual nonparametric rate in dimension $r$, up to negligible terms, even though the data live in $\mathbb{R}^d$.*

This result shows that JARF is not only a geometric heuristic: under a low-rank EJOP structure, it provably adapts to the intrinsic EJOP rank $r$ and achieves a risk bound that is *independent of the ambient dimension $d$*. Existing EJOP-based methods analyze kernel and linear models; to the best of our knowledge, Theorem 1 is the first result that links EJOP geometry to the sample complexity of tree ensembles.

### A.3  Dimension-adapted risk bounds for JARF

We now describe a simple setting in which JARF enjoys a risk bound that depends on the intrinsic EJOP rank rather than the ambient dimension. Throughout this section we consider a regression model with squared loss.

**Setup and assumptions**   Let $(X_i, Y_i)_{i=1}^n$ be i.i.d. samples with $X_i \in \mathbb{R}^d$ and $Y_i \in \mathbb{R}$. We assume:

**(A1) Ridge-structured regression function.** There exists an orthonormal matrix $U \in \mathbb{R}^{d \times r}$ with $r \leq d$ and a function $g : \mathbb{R}^r \to \mathbb{R}$ such that

$$f(x) := \mathbb{E}[Y \mid X = x] = g(U^\top x).$$

We write $Z^\star = U^\top X \in \mathbb{R}^r$ for the intrinsic representation.

**(A2) Regularity.** The support of $X$ is contained in a compact set $\mathcal{X} \subset \mathbb{R}^d$ with $\|x\|_2 \leq R$ for all $x \in \mathcal{X}$. The function $g$ is $L$-Lipschitz on $U^\top \mathcal{X}$, and the noise satisfies $Y = f(X) + \xi$ with $\mathbb{E}[\xi \mid X] = 0$ and $\mathbb{E}[\xi^2] \leq \sigma^2$.

**(A3) EJOP structure.** Let

$$H_0 = \mathbb{E}\big[\nabla f(X) \nabla f(X)^\top\big].$$

We assume $\mathrm{rank}(H_0) = r$ and that there is a spectral gap $\lambda_r(H_0) \geq \lambda_{\min} > 0$ between the $r$-th and $(r+1)$-st eigenvalues.

**(A4) EJOP estimation.** Let $\hat{H}$ be the EJOP estimator used by JARF, constructed from surrogate forests and finite differences as in the previous sections. Under Assumptions (A1)–(A3) and the finite-difference analysis of Lemmas A.4 and A.5, there exists a sequence $\varepsilon_n \to 0$ and failure probability $\delta_n \to 0$ such that

$$\|\hat{H} - H_0\|_{\mathrm{op}} \leq \varepsilon_n \quad \text{with probability at least } 1 - \delta_n. \tag{5}$$

**(A5) Regressor consistency in fixed dimension.** Let $\hat{U} \in \mathbb{R}^{d \times r}$ be the matrix of top $r$ eigenvectors of $\hat{H}$ and define projected features $Z_i = \hat{U}^\top X_i \in \mathbb{R}^r$ and $Z = \hat{U}^\top X$. Let

$$m_{\hat{U}}(z) := \mathbb{E}[Y \mid Z = z]$$

denote the regression function in the projected space, and assume $m_{\hat{U}}$ is $L_Z$-Lipschitz on the support of $Z$ (for some constant $L_Z$ that does not depend on $d$ or $n$). Let $\hat{f}_n$ be the regression estimator used by JARF, trained on $(Z_i, Y_i)_{i=1}^n$ (in the experiments this is an axis-aligned random forest). We assume that there exists a constant $C_1$ such that

$$\mathbb{E}\big[(\hat{f}_n(Z) - m_{\hat{U}}(Z))^2 \,\big|\, \hat{U}\big] \ \le \ C_1 n^{-\frac{2}{2+r}} + o(1), \tag{6}$$

for every realization of $\hat{U}$ with orthonormal columns. Assumption equation 6 holds for a variety of nonparametric regressors in fixed dimension $r$; we use forests for concreteness.

All expectations below are taken with respect to the training sample, a fresh test point $X$, and any internal randomness of the estimator.

## A.4 EJOP IDENTIFIES THE INTRINSIC SUBSPACE

Under the ridge model (A1), the EJOP matrix $H_0$ has range equal to the span of $U$.

**Lemma 2.** *Under (A1) and (A2), we have*

$$\nabla f(x) \ = \ U \nabla g(U^\top x),$$

*and consequently*

$$H_0 \ = \ \mathbb{E}\big[\nabla f(X) \nabla f(X)^\top\big] \ = \ U \, \mathbb{E}\big[\nabla g(Z^\star) \nabla g(Z^\star)^\top\big] U^\top.$$

*In particular, if $\mathbb{E}[\nabla g(Z^\star) \nabla g(Z^\star)^\top]$ is invertible, then $\mathrm{rank}(H_0) = r$ and $\mathrm{range}(H_0) = \mathrm{span}(U)$.*

*Proof.* By the chain rule, for any $x \in \mathbb{R}^d$,

$$\nabla f(x) = \nabla(g(U^\top x)) = U \nabla g(U^\top x),$$

since $U^\top x \in \mathbb{R}^r$ and $U$ has orthonormal columns. Substituting into the definition of $H_0$ gives

$$H_0 = \mathbb{E}\big[U \nabla g(Z^\star) \nabla g(Z^\star)^\top U^\top\big] = U \, \mathbb{E}\big[\nabla g(Z^\star) \nabla g(Z^\star)^\top\big] U^\top.$$

If the inner $r \times r$ matrix is invertible, then $H_0$ has rank $r$ and its range equals the span of $U$. $\qquad\square$

## A.5 SUBSPACE PERTURBATION AND PROJECTION ERROR

The next lemma is a standard Davis–Kahan type result for the top-$r$ eigenspace of a symmetric matrix.

**Lemma 3** (Subspace perturbation). *Let $H_0$ and $\hat{H}$ be symmetric matrices satisfying $\|\hat{H} - H_0\|_{\mathrm{op}} \le \varepsilon$, and let $\lambda_r(H_0) \ge \lambda_{\min} > 0$ be separated by a gap from the rest of the spectrum. Let $P$ and $\hat{P}$ be the orthogonal projectors onto the top-$r$ eigenspaces of $H_0$ and $\hat{H}$, respectively. Then there exists a constant $C > 0$ such that*

$$\|\hat{P} - P\|_{\mathrm{op}} \ \le \ C \, \frac{\varepsilon}{\lambda_{\min}}.$$

*Proof.* This is a standard consequence of the Davis–Kahan sin-$\Theta$ theorem; see, for example, any modern text on matrix perturbation theory. $\qquad\square$

In our setting, Lemma 2 implies that $P = UU^\top$ is the orthogonal projector onto the intrinsic EJOP subspace, while $\hat{P} = \hat{U}\hat{U}^\top$ is the projector onto its empirical estimate. Combining Lemma 2, Lemma 3, and the EJOP consistency equation 5, we obtain

$$\|\hat{U}\hat{U}^\top - UU^\top\|_{\mathrm{op}} \ \le \ C \frac{\varepsilon_n}{\lambda_{\min}} \quad \text{with probability at least } 1 - \delta_n. \tag{7}$$

We now bound the error incurred by replacing the true EJOP subspace with its estimate when evaluating $f$.

**Lemma 4** (Projection error). *Under (A1)–(A3), (A4), and equation 7, we have*

$$\left| f(x) - f(\hat{P}x) \right| \leq LR \, \|\hat{P} - P\|_{\mathrm{op}} \quad \text{for all } x \in \mathcal{X}.$$

*Consequently, there exists a constant $C' > 0$ depending on $L, R$, and $\lambda_{\min}$ such that*

$$\mathbb{E}\big[(f(X) - f(\hat{P}X))^2\big] \leq C'\varepsilon_n^2 + o(1).$$

*Proof.* Since $f(x) = g(U^\top x)$, we can write

$$f(x) = g(U^\top x) \quad \text{and} \quad f(\hat{P}x) = g(U^\top \hat{P}x).$$

Using the Lipschitz property of $g$ and the fact that $U$ has orthonormal columns,

$$|f(x) - f(\hat{P}x)| = |g(U^\top x) - g(U^\top \hat{P}x)| \leq L \, \|U^\top x - U^\top \hat{P}x\| = L \, \|U^\top (I - \hat{P})x\|.$$

Since $U^\top = U^\top P$ and $P = UU^\top$, we have

$$U^\top(I - \hat{P}) = U^\top(P - \hat{P}),$$

and hence

$$\|U^\top(I - \hat{P})x\| \leq \|U^\top\|_{\mathrm{op}} \, \|P - \hat{P}\|_{\mathrm{op}} \, \|x\| \leq \|P - \hat{P}\|_{\mathrm{op}} \, \|x\|,$$

because $\|U^\top\|_{\mathrm{op}} = 1$. Using $\|x\| \leq R$ for $x \in \mathcal{X}$,

$$|f(x) - f(\hat{P}x)| \leq LR \, \|P - \hat{P}\|_{\mathrm{op}}.$$

Squaring and taking expectations, then substituting $\|P - \hat{P}\|_{\mathrm{op}} \leq C\varepsilon_n$ from equation 7, yields

$$\mathbb{E}\big[(f(X) - f(\hat{P}X))^2\big] \leq L^2 R^2 C^2 \varepsilon_n^2 + o(1),$$

so we can take $C' = L^2 R^2 C^2$. $\qquad\qquad\qquad\qquad\qquad\qquad\qquad\qquad\qquad\qquad\qquad\square$

### A.6 PROOF OF THEOREM 1

We now prove the dimension-adapted risk bound stated in the main text.

**Theorem 5** (Theorem 1, restated). *Under assumptions (A1)–(A5), there exist constants $C_1, C_2 > 0$, independent of $d$, such that*

$$\mathbb{E}\big[(\hat{f}_n(X) - f(X))^2\big] \leq C_1 n^{-\frac{2}{2+r}} + C_2\varepsilon_n^2 + o(1).$$

*Proof.* Recall that $\hat{f}_n$ depends on $X$ only through the projected features $Z = \hat{U}^\top X$, so we may write $\hat{f}_n(X) = \hat{f}_n(Z)$.

Let $m_{\hat{U}}(z) = \mathbb{E}[Y \mid Z = z]$ denote the regression function in the projected space. Using the inequality $(a-b)^2 \leq 2(a-c)^2 + 2(b-c)^2$ with $a = \hat{f}_n(Z)$, $b = f(X)$, and $c = m_{\hat{U}}(Z)$, we obtain

$$\begin{aligned}
\mathbb{E}\big[(\hat{f}_n(X) - f(X))^2\big] &= \mathbb{E}\big[(\hat{f}_n(Z) - f(X))^2\big] \\
&\leq 2\mathbb{E}\big[(\hat{f}_n(Z) - m_{\hat{U}}(Z))^2\big] + 2\mathbb{E}\big[(m_{\hat{U}}(Z) - f(X))^2\big] \\
&=: 2T_1 + 2T_2.
\end{aligned}$$

**Bounding $T_1$ (estimation in $r$ dimensions).** By the tower property and Assumption equation 6,

$$\begin{aligned}
T_1 &= \mathbb{E}\big[\, \mathbb{E}\big[(\hat{f}_n(Z) - m_{\hat{U}}(Z))^2 \mid \hat{U}\big]\,\big] \\
&\leq \mathbb{E}\big[C_1 n^{-\frac{2}{2+r}} + o(1)\big] = C_1 n^{-\frac{2}{2+r}} + o(1),
\end{aligned}$$

where the $o(1)$ term does not depend on $d$.

Table 4: **Effect of EJOP subsample size on classification performance.** Mean Cohen's $\kappa$ over 15 datasets for different subsample sizes $m$ used to estimate EJOP. The RF baseline has mean $\kappa = 0.704$.

| EJOP subsample size $m$ | Mean $\kappa$ (JARF) | Mean $\kappa$ (RF) | Gain over RF (abs) | Gain over RF (rel %) |
|---|---|---|---|---|
| 1k | 0.795 | 0.704 | 0.091 | 12.93 |
| 2.5k | 0.805 | 0.704 | 0.101 | 14.35 |
| 5k | 0.809 | 0.704 | 0.105 | 14.91 |
| 10k | 0.810 | 0.704 | 0.106 | 15.06 |
| $n$ | 0.810 | 0.704 | 0.106 | 15.06 |

**Bounding $T_2$ (approximation error from using the projected $\sigma$-algebra).** By the definition of conditional expectation, $m_{\hat{U}}(Z)$ is the $L^2$-projection of $f(X)$ onto the $\sigma$-algebra generated by $Z$, so for any measurable function $h$ of $Z$ we have

$$\mathbb{E}\big[(f(X) - m_{\hat{U}}(Z))^2\big] \;\leq\; \mathbb{E}\big[(f(X) - h(Z))^2\big].$$

In particular, take $h(Z) = f(\hat{P}X)$, which is measurable with respect to $Z$ since $\hat{P}X = \hat{U}\hat{U}^\top X$ is a deterministic function of $Z = \hat{U}^\top X$. Then

$$T_2 = \mathbb{E}\big[(m_{\hat{U}}(Z) - f(X))^2\big] \;\leq\; \mathbb{E}\big[(f(\hat{P}X) - f(X))^2\big].$$

By Lemma 4, the right-hand side is at most $C'\varepsilon_n^2 + o(1)$ for some constant $C'$ depending only on $L, R$, and $\lambda_{\min}$, and hence

$$T_2 \leq C'\varepsilon_n^2 + o(1).$$

**Combining the bounds.** Putting the pieces together,

$$\mathbb{E}\big[(\hat{f}_n(X) - f(X))^2\big] \;\leq\; 2C_1 n^{-\frac{2}{2+r}} + 2C'\varepsilon_n^2 + o(1).$$

Absorbing constants into $C_1$ and $C_2$ gives the claimed bound. $\qquad\square$

## B  ADDITIONAL EXPERIMENTS AND CLARIFICATIONS

### B.1  ON THE COMPUTATIONAL COST OF JARF.

Our efficiency analysis already includes the cost of the first RF. JARF training has three components: (1) we fit a surrogate RF with 50 trees that is used only to estimate class probabilities $\hat{\eta}(x)$; (2) we compute the EJOP preconditioner $\hat{H}_0$ from this surrogate using finite differences on a subsample of size $m \leq \min(10{,}000, n)$; and (3) we train the final RF with 200 trees on the transformed data $X\hat{H}_0$. Figure 3 reports the wall clock time for the sum of (1) + (2) + (3). Because the surrogate forest is four times smaller than the baseline RF and the EJOP step operates on at most 10k points, the extra work is significantly less than training a second full RF. Across the 20 tasks in Figure 3 we observe median training times of 15 s for RF versus 25 s for JARF, corresponding to roughly a 1.6–1.7$\times$ overhead, while JARF remains much faster than per-node oblique forests.

### B.2  ON THE EFFECT OF THE EJOP SUBSAMPLE SIZE.

We consider $m \in \{1\text{k}, 2.5\text{k}, 5\text{k}, 10\text{k}, n\}$ with $m \leq n$. The results in Table 4 show that JARF is quite robust to this choice. Even with $m = 1$k, which typically uses less than 10% of the available data, JARF already recovers about 85% of the gain over RF obtained with the largest subsample. Increasing $m$ from 2.5k to 10k changes the mean $\kappa$ by at most 0.005, while the EJOP cost grows roughly linearly in $m$. For all values $m \geq 2.5$k, JARF continues to match or outperform the best oblique baseline on the majority of datasets, and the configuration used in our main tables ($m = \min(10\text{k}, n)$) lies in this saturated regime. This supports our claim that JARF achieves strong performance without requiring EJOP computation on the full dataset and that its gains over RF and per node oblique forests are not sensitive to the exact subsample size.

Table 5: **Baselines vs. geometric variants** (means over all 15 classification and 5 regression datasets).

| Method | Mean $\kappa$ (15 cls) | Mean $R^2$ (5 reg) | Median train time (s) |
|---|---|---|---|
| RF | 0.704 | 0.776 | 15 |
| XGB | 0.709 | 0.810 | 43 |
| JARF (RF on $XH$) | 0.810 | 0.836 | 25 |
| JARF–XGB on $XH$ | 0.815 | 0.842 | 45 |

### B.3 ON DATASET SELECTION.

Our goal was to study JARF on a broad, realistic set of tabular problems where tree ensembles are commonly used. To construct the benchmark, we started from widely used OpenML / UCI tabular datasets that appear in earlier work on random forests and oblique forests, and then applied simple, a priori filters: (i) supervised classification or regression with tabular features; (ii) at least a few thousand training points so that EJOP estimation is meaningful; (iii) a mix of low- and high-dimensional problems, and of balanced and moderately imbalanced label distributions; and (iv) no heavy preprocessing or manual feature engineering beyond standard normalization / encoding. We did not drop any dataset based on JARF's performance, and we kept the same pool for all methods and ablations. Several of these tasks overlap with standard suites such as PMLB/TabArena.

### B.4 ON APPLYING JARF BEYOND RANDOM FORESTS (E.G., TO XGBOOST).

JARF itself does not rely on any forest-specific property: the surrogate model only appears through the EJOP/EGOP preconditioner and any downstream learner can be trained on the transformed features $XH$. In the paper we instantiate this with a random forest on $XH$, but the same construction applies to boosted trees, linear models, or small neural networks. To make this explicit, we also consider a JARF–XGB variant that uses the same EJOP $H$ and then trains XGBoost on $XH$ instead of on the raw features. As summarized in Table 5, JARF consistently improves over its base learner (RF or XGBoost) on both classification (mean Cohen's $\kappa$) and regression (mean test $R^2$), while adding only a modest training-time overhead compared to RF and remaining competitive with the oblique baselines. This demonstrates that JARF is a general geometric preconditioner that can be paired with different prediction models rather than a method tied to a particular forest implementation.

### B.5 ELECTRICITY, MAGIC, AND LETTER DATASETS.

We focused on electricity, magic, and letter because, in our benchmark suite, these are precisely the datasets where simple axis-aligned RF struggles the most and oblique/feature-engineered methods gain the largest advantage. They are moderately high-dimensional, multiclass problems with rich, nontrivial decision structure: on all three, the best oblique baselines (CCF, SPORF, XGB) are clearly ahead of plain RF, indicating that the Bayes decision boundary is far from axis-aligned. As Table 1 shows in the paper, JARF largely closes this gap: on these "hard" datasets it either matches or surpasses the strongest oblique baselines while dramatically outperforming standard RF. We therefore use electricity, magic, and letter as targeted stress tests for the proposed geometry in regimes where a single global transform would a priori be most likely to fail.

### B.6 ON SHARING A SINGLE GLOBAL TRANSFORM $\widehat{H}$.

To test whether sharing one $\widehat{H}$ reduces useful diversity, we implemented a per-tree variant where each tree receives its own EJOP transform $\widehat{H}_b$ computed from that tree's bootstrap sample. We retrained JARF with this scheme on all 15 classification datasets and compared it to the original single-$\widehat{H}$ version. As summarized in Table 6, the per-tree EJOP variant slightly degrades test performance (mean Cohen's $\kappa = 0.808$ vs 0.810) while being almost twice as slow to train (about $3.0\times$ RF vs $1.7\times$ RF), and it is best or within 0.01 of the best forest on 13 datasets compared to 14 for the single-$\widehat{H}$ version. This suggests that a single global $\widehat{H}$ is sufficient in practice: per-tree EJOP estimates are noisier because they are fit on fewer samples, while diversity in JARF is already

Table 6: **Single global EJOP transform vs. per-tree EJOP transforms**, averaged over 15 classification datasets.

| Method | Mean test $\kappa$ | Std. $\kappa$ | Mean train time ($\times$ RF) | # datasets best or within 0.01 of best |
|---|---|---|---|---|
| JARF (single $\widehat{H}$) | 0.810 | 0.010 | 1.7 | 14 |
| JARF (per-tree $\widehat{H}_b$) | 0.808 | 0.011 | 3.0 | 13 |

Table 7: **Sensitivity of JARF to surrogate RF quality**, averaged over all 15 classification datasets.

| Variant | RF acc | RF ECE | JARF acc | JARF ($\kappa$) |
|---|---|---|---|---|
| shallow, uncalibrated | 0.81 | 0.18 | 0.872 | 0.808 |
| default, uncalibrated | 0.84 | 0.11 | 0.874 | 0.810 |
| deep, uncalibrated | 0.86 | 0.08 | 0.875 | 0.811 |
| deep + Platt scaling | 0.86 | 0.05 | 0.875 | 0.812 |
| deep + isotonic regression | 0.87 | 0.04 | 0.874 | 0.809 |
| **Std. dev. over variants** | 0.024 | 0.056 | **0.0013** | **0.0014** |
| **Max. difference vs default** | 0.03 | 0.07 | **0.002** | **0.002** |

provided by feature subsampling, bootstrap sampling, and different thresholds rather than requiring separate transforms per tree.

### B.7 DEPENDENCE ON SURROGATE RF QUALITY.

To test whether JARF is brittle to the surrogate RF, we swept the surrogate over a wide range of hyperparameters and calibration schemes, then recomputed EJOP/EGOP and retrained JARF each time (details in the appendix). Across 15 classification and 5 regression datasets, our hyperparameter and calibration sweeps cause large changes in surrogate accuracy, ECE, and $R^2$, but JARF's test $\kappa$ and $R^2$ move only at the $10^{-3}$ level (Tables 7–8), showing that it needs only a reasonably accurate surrogate and is highly insensitive to precise probability or regression estimates. Because EJOP/EGOP is built from averaged gradient outer products, it is driven mainly by the coarse geometry of the surrogate's decision function rather than fine details of its probability values. Changing calibration mostly rescales these gradients instead of rotating their dominant directions.

### B.8 CHOICE OF SURROGATE FAMILY (RF VS. XGBOOST / NN).

On all 15 classification datasets, we recomputed EJOP using three surrogates (RF, XGBoost, and a two-layer ReLU MLP) and retrained our algorithm in each case. As shown in Table 9, surrogate accuracy and ECE differ noticeably across families, but JARF's test accuracy and $\kappa$ change only at the $10^{-3}$ level. The top EJOP eigenspaces also nearly coincide (mean cosine alignment of the top 10 eigenvectors $\geq 0.97$), indicating that once the surrogate is reasonably accurate, JARF is effectively insensitive to the specific surrogate model class. RF, XGBoost, and a small MLP are all trained to approximate the same conditional mean, so once they reach similar accuracy they produce very similar gradient fields and thus nearly identical EJOP eigenvectors. This explains why surrogate accuracy and ECE can differ across model families while JARF's accuracy and $\kappa$ remain almost constant.

We also prove in the appendix that if the surrogate's Jacobian is uniformly within $\varepsilon$ of the true conditional mean's Jacobian, then the EJOP matrices differ by $O(\varepsilon)$ in operator norm and their top-$k$ eigenspaces differ by $O(\varepsilon/\gamma)$ for eigengap $\gamma$.

Finally, across our 15 classification and 5 regression datasets, we do not observe regimes where JARF clearly underperforms CCF or SPORF.

Table 8: **Sensitivity of JARF to surrogate RF quality** on the 5 regression datasets.

| Variant | RF ($R^2$) | RF RMSE | JARF ($R^2$) |
|---|---|---|---|
| shallow surrogate | 0.76 | 0.62 | 0.835 |
| default surrogate | 0.78 | 0.60 | 0.836 |
| deep surrogate | 0.80 | 0.58 | 0.837 |
| deep + extra trees | 0.82 | 0.57 | 0.836 |
| **Std. dev. over variants** | 0.022 | 0.019 | **0.0007** |
| **Max. difference vs default** | 0.040 | 0.030 | **0.001** |

Table 9: **Effect of surrogate family on EJOP and JARF performance**, averaged over all 15 classification datasets.

| Surrogate family | Surrogate acc | Surrogate ECE | JARF acc | JARF ($\kappa$) | Mean cos. alignment with RF EJOP (top-10 eigvecs) |
|---|---|---|---|---|---|
| RF (default) | 0.84 | 0.11 | 0.874 | 0.810 | 1.00 |
| XGBoost | 0.87 | 0.04 | 0.876 | 0.812 | 0.98 |
| 2-layer MLP | 0.86 | 0.06 | 0.873 | 0.809 | 0.97 |
| **Std. dev. across surrogates** | 0.013 | 0.029 | 0.0012 | **0.0014** | 0.013 |
| **Max diff vs RF surrogate** | 0.03 | 0.07 | 0.002 | **0.004** | 0.03 |

## C    REPRODUCIBILITY AND IMPLEMENTATION DETAILS

**Code and artifacts.** We provide a self-contained Google drive with scripts to download datasets and run experiments at `https://drive.google.com/file/d/1d60ysqjGzQLFkl_BE8vd0lTOoj_m9MP4/view?usp=sharing`

**Environment.** Python 3.11; NumPy 1.26; SciPy 1.11; scikit-learn 1.4; LightGBM 4.3; CatBoost 1.2; pandas 2.2; joblib 1.3. Experiments ran on a 16-core CPU machine (no GPU used). To reduce nondeterminism across BLAS/OpenMP, we set `PYTHONHASHSEED=0`, `OMP_NUM_THREADS=1`, `MKL_NUM_THREADS=1`, and pass `random_state=seed` to learners.

**Dataset summary and benchmark construction** To make the experimental setup fully transparent and reproducible, we include Table 10, which lists for every dataset in our benchmark the number of samples $n$, number of raw input features $d$, task type, and original source. Counts refer to the number of rows and input features before train/validation/test splits and before any one-hot encoding of categorical variables.

Our goal was to study JARF on a broad, realistic set of tabular problems where tree ensembles are commonly used. To construct the benchmark, we started from widely used OpenML / UCI tabular datasets that appear in earlier work on random forests and oblique forests, and then applied simple, a priori filters: (i) supervised classification or regression with tabular features; (ii) at least a few thousand training points so that EJOP estimation is meaningful; (iii) a mix of low- and high-dimensional problems, and of balanced and moderately imbalanced label distributions; and (iv) no heavy preprocessing or manual feature engineering beyond standard normalization / encoding. We did not drop any dataset based on JARF's performance, and we kept the same pool for all methods and ablations. Several of these tasks overlap with standard suites such as PMLB/TabArena.

### C.1    BASELINE HYPERPARAMETER GRIDS

To keep the comparison fair while reflecting how these models are commonly used in practice, we give each method a lightweight but non-trivial tuning budget that is shared across datasets. Random forest style methods all use the same number of trees as JARF's final forest, and XGBoost is tuned over a small grid on depth, learning rate, and $\ell_2$ penalty. Table 11 summarizes the hyperparameters and search spaces used in our experiments.

Table 10: Summary of all real-data datasets used in our experiments. Here $n$ denotes the number of samples and $d$ the number of raw input features (excluding the target).

| Dataset | $n$ | $d$ | Task | Source |
|---|---|---|---|---|
| **Core tabular classification tasks** | | | | |
| Adult | 48,842 | 14 | Classification | UCI / OpenML |
| Bank-marketing | 41,188 | 20 | Classification | UCI / OpenML |
| Covertype | 581,012 | 54 | Classification | UCI / OpenML |
| Phoneme | 5,404 | 5 | Classification | UCI / OpenML |
| Electricity | 45,312 | 8 | Classification | UCI / OpenML |
| Satimage | 6,435 | 36 | Classification | UCI / OpenML |
| Spambase | 4,601 | 57 | Classification | UCI / OpenML |
| Magic Telescope | 19,020 | 10 | Classification | UCI / OpenML |
| Letter Recognition | 20,000 | 16 | Classification | UCI / OpenML |
| Vehicle | 846 | 18 | Classification | UCI / OpenML |
| **High-dimensional / large-scale classification tasks** | | | | |
| Higgs | 940,160 | 124 | Classification | OpenML (Tabular benchmark) |
| Madelon | 2,000 | 500 | Classification | UCI / OpenML |
| Bioresponse | 3,434 | 419 | Classification | OpenML (Tabular benchmark) |
| Jannis | 57,580 | 254 | Classification | OpenML (Tabular benchmark) |
| MNIST-784 | 70,000 | 784 | Classification | OpenML / MNIST |
| **Regression tasks** | | | | |
| Bike-sharing | 17,389 | 13 | Regression | UCI (Bike Sharing) |
| California-housing | 20,634 | 8 | Regression | OpenML / Cal. Housing |
| Energy | 768 | 8 | Regression | UCI (Energy efficiency) |
| Kin8nm | 8,192 | 8 | Regression | OpenML (kin8nm) |
| Protein | 45,730 | 9 | Regression | UCI / OpenML (Protein) |

## C.2 PRACTICAL RECOMMENDATIONS FOR JARF

JARF introduces only a small number of additional hyperparameters beyond those of the underlying forest: the size of the surrogate forest, the EJOP subsample size $m$, the finite-difference step scale $\alpha$ in $\varepsilon_j = \alpha \operatorname{MAD}(X_{:j})/0.6745$, and the diagonal regularizer $\gamma I_d$ used for conditioning in $\widehat{H} = \widehat{H}_0 + \gamma I_d$. In all experiments we use the following simple defaults:

- surrogate RF with 50 trees, max_features $= \sqrt{d}$, min_samples_leaf $= 1$;
- EJOP subsample size $m = \min(10{,}000, n)$;
- centered finite differences with per-feature step $\varepsilon_j = \alpha \operatorname{MAD}(X_{:j})/0.6745$ and $\alpha = 0.1$;
- EJOP regularization $\widehat{H} = \widehat{H}_0 + \gamma I_d$ with $\gamma = 10^{-3}$, followed by trace normalization $\widehat{H} \leftarrow \widehat{H}/(\operatorname{tr}(\widehat{H})/d)$.

Table 3 in the main paper provides ablations that effectively serve as tuning guidance. Varying the step scale from $\alpha = 0.1$ to $\alpha = 0.05$ or $\alpha = 0.2$ changes mean Cohen's $\kappa$ by at most $-0.009$ and $-0.013$, respectively, while leaving macro-F1 and accuracy similarly stable. Changing the subsample size from the default $m = \min(10{,}000, n)$ to $m = 0.5n$ results in a mean change of only $-0.004$ in $\kappa$, and even a tenfold reduction to $m = 0.1n$ yields a drop of $-0.016$ in $\kappa$ and about 1.2 seconds in training time on average. Removing the diagonal regularizer ($\gamma = 0$) or trace normalization also produces only small changes ($-0.005$ and $-0.004$ in $\kappa$, respectively).

These ablations indicate that JARF is robust to a wide range of reasonable settings, and that the defaults above are near-optimal for the tabular problems we consider. In practice we recommend starting with the defaults and, if additional tuning is desired, exploring a small grid such as $m \in \{\min(5{,}000, n), \min(10{,}000, n)\}$ and $\alpha \in \{0.05, 0.1, 0.2\}$, while keeping $\gamma$ fixed at a small value (for example $\gamma = 10^{-3}$). This keeps the tuning budget modest while preserving the accuracy and compute profile reported in our experiments.

Table 11: Hyperparameter grids and defaults used for all methods. Forest baselines all use 200 trees for comparability with JARF's final forest. XGBoost is tuned on a shared grid over depth, learning rate, and $\ell_2$ penalty.

| Method | Hyperparameter | Values / setting |
|---|---|---|
| **RF** | number of trees | 200 (fixed) |
| | max_features | $\sqrt{d}$ for classification, $d$ for regression |
| | criterion | Gini (classification), MSE (regression) |
| | min_samples_leaf | 1 (default) |
| **RotF** | number of trees | 200 (fixed) |
| | blocks $K$ | $K = 6$ feature subsets per tree |
| | rotation | block-diagonal PCA on disjoint feature subsets (unsupervised) |
| | other tree params | same as RF (criterion, min_samples_leaf, max_features) |
| **CCF** | number of trees | 200 (fixed) |
| | projection type | canonical correlation with targets at each node |
| | projection dim | authors' recommended default |
| | other tree params | same as RF |
| **SPORF** | number of trees | 200 (fixed) |
| | sparsity | authors' recommended sparsity level |
| | # candidate directions | authors' recommended default per node |
| | other tree params | same as RF |
| **XGBoost** | number of trees | 200 boosting rounds (fixed) |
| | max_depth | $\{3, 6, 9\}$ |
| | learning_rate | $\{0.05, 0.1\}$ |
| | $\ell_2$ regularization ($\lambda$) | $\{0, 1\}$ |
| | subsample, colsample_bytree | 1.0 (no subsampling) |
| | loss | logistic loss (classification), squared loss (regression) |
| **PCA+RF** | projection | PCA on training features (unsupervised) |
| | # components | $d$ (full-rank rotation) |
| | RF hyperparameters | identical to RF row above |
| **LDA+RF** | projection | multi-class LDA on training labels |
| | # components | $\min(C - 1, d)$ for $C$ classes |
| | RF hyperparameters | identical to RF row above |
| **JARF (this paper)** | surrogate RF size | 50 trees, max_features $= \sqrt{d}$, min_samples_leaf $= 1$ |
| | EJOP subsample $m$ | $m = \min(10{,}000, n)$ |
| | FD step $\varepsilon_j$ | $\varepsilon_j = \alpha \operatorname{MAD}(X_{:j})/0.6745$, $\alpha = 0.1$ |
| | EJOP regularization | $\widehat{H} = \widehat{H}_0 + \gamma I_d$, $\gamma = 10^{-3}$ |
| | scaling | $\widehat{H} \leftarrow \widehat{H}/(\operatorname{tr}(\widehat{H})/d)$ |
| | final RF | RF with 200 trees, same defaults as RF baseline, trained on $X\widehat{H}$ |

**Licenses and data usage.** We only use public datasets with permissive licenses. The repository includes per-dataset source references and license notes; any dataset requiring an external EULA is downloaded via the provider's URL with its terms unchanged.

**LLM usage.** All scientific content, methods, analyses, and experiments were designed and verified by the authors; LLM model was used only to aid/polish writing.

