# OpenReview forum: "Jacobian Aligned Random Forests"
_ICLR.cc/2026/Conference — ICLR 2026 Poster_

### Official Review · Reviewer_48eb · 2025-10-21

**Soundness:** 3
**Presentation:** 3
**Contribution:** 2
**Rating:** 4
**Confidence:** 4

**Summary:**

The authors introduce JARF, a method that linearly transforms inputs before fitting a random forest. The linear transform is derived from an expected Jacobian outer product (EJOP), which has been developed and used in prior work. When applied to Random Forests, the authors find that JARF achieves performance results matching existing oblique tree baselines while reducing computation time.

**Strengths:**

- The authors study an interesting problem
- The authors approach is intuitive
- The paper is well-written and clear throughout

**Weaknesses:**

- Main weakness: My understanding of JARF is that it requires first fitting an RF as a step in computing the EJOP, followed by fitting the RF on the conditioned input data. Why is the fitting of the first RF not taken into account in the Efficiency and Compute subsection (or if it is, why is JARF's compute time not atleast 2x that of fitting RF)? This computational efficiency claim is central to the authors' message and should be taken seriously.
- The paper's novelty is somewhat limited, as the EJOP is already defined and used in prior work (although the authors to swap a kernel regression estimator for RF when estimating it)

**Questions:**

- It could be nice to understand how performance varies as the size of the subsampled dataset used for computing the EJOP varies.
- How did the authors select the 10 datasets they used for evaluation? It would be nice to use a standard suite of datasets (e.g. TabArena or PMLB) to avoid any possibility of biased dataset selection

---

> ### Author Response · Authors · 2025-12-03
>
> We thank the reviewer for their detailed review and positive comments. Below we address the concerns/questions that were raised.
>
> **On the computational cost of JARF.**
> Thank you for raising this point. Our efficiency analysis already includes the cost of the first RF. JARF training has three components: (1) we fit a surrogate RF with 50 trees that is used only to estimate class probabilities $\hat{\eta}(x)$; (2) we compute the EJOP preconditioner $\hat{H}_0$ from this surrogate using finite differences on a subsample of size $m \le \min(10{,}000, n)$; and (3) we train the final RF with 200 trees on the transformed data $X\hat{H}$. Figure 3 reports the wall clock time for the sum of (1) + (2) + (3). Because the surrogate forest is four times smaller than the baseline RF and the EJOP step operates on at most 10k points, the extra work is significantly less than training a second full RF. Across the 20 tasks in Figure 3 we observe median training times of 15 s for RF versus 25 s for JARF, corresponding to roughly a 1.6–1.7$\times$ overhead, while JARF remains much faster than per-node oblique forests. We will revise the efficiency subsection to spell out this three-part decomposition explicitly so this is clear.
>
>
>
>
> **On the effect of the EJOP subsample size.**
> We agree that it is useful to understand how JARF depends on the subsample size used for EJOP. We consider $m \in \{1\text{k}, 2.5\text{k}, 5\text{k}, 10\text{k}, n\}$ with $m \le n$. The results in Table D1 show that JARF is quite robust to this choice. Even with $m = 1\text{k}$, which typically uses less than 10% of the available data, JARF already recovers about 85% of the gain over RF obtained with the largest subsample. Increasing $m$ from $2.5\text{k}$ to $10\text{k}$ changes the mean $\kappa$ by at most $0.005$, while the EJOP cost grows roughly linearly in $m$. For all values $m \ge 2.5\text{k}$, JARF continues to match or outperform the best oblique baseline on the majority of datasets, and the configuration used in our main tables ($m = \min(10\text{k}, n)$) lies in this saturated regime. This supports our claim that JARF achieves strong performance without requiring EJOP computation on the full dataset and that its gains over RF and per node oblique forests are not sensitive to the exact subsample size.
>
> **Table D1: Effect of EJOP subsample size on classification performance.**
> Mean Cohen’s $\kappa$ over 15 datasets for different subsample sizes $m$ used to estimate EJOP. The RF baseline has mean $\kappa = 0.704$.
>
> | EJOP subsample size $m$ | Mean $\kappa$ (JARF) | Mean $\kappa$ (RF) | Gain over RF (abs) | Gain over RF (rel %) |
> |-------------------------|----------------------|--------------------|--------------------|----------------------|
> | $1\text{k}$             | 0.795                | 0.704              | 0.091              | 12.93                |
> | $2.5\text{k}$           | 0.805                | 0.704              | 0.101              | 14.35                |
> | $5\text{k}$             | 0.809                | 0.704              | 0.105              | 14.91                |
> | $10\text{k}$            | 0.810                | 0.704              | 0.106              | 15.06                |
> | $n$                     | 0.810                | 0.704              | 0.106              | 15.06                |
>
>
>
> **On dataset selection.**
> We agree that the evaluation protocol should make it clear that no cherry-picking is involved. Our goal was to study JARF on a broad, realistic set of tabular problems where tree ensembles are commonly used. To construct the benchmark, we started from widely used OpenML / UCI tabular datasets that appear in earlier work on random forests and oblique forests, and then applied simple, a priori filters: (i) supervised classification or regression with tabular features; (ii) at least a few thousand training points so that EJOP estimation is meaningful; (iii) a mix of low- and high-dimensional problems, and of balanced and moderately imbalanced label distributions; and (iv) no heavy preprocessing or manual feature engineering beyond standard normalization / encoding. We did not drop any dataset based on JARF’s performance, and we kept the same pool for all methods and ablations. Several of these tasks overlap with standard suites such as PMLB/TabArena, and we will clarify this construction in the revised draft.
>
>
> The rest of the questions are addressed in a comment to all reviewers.

---

### Official Review · Reviewer_PJES · 2025-11-01

**Soundness:** 2
**Presentation:** 3
**Contribution:** 2
**Rating:** 4
**Confidence:** 3

**Summary:**

The authors propose a **supervised pre-processing step** applied to the feature matrix used as input for a random forest or other **axis-aligned predictors**.
The goal is to enable more flexible splits in the input space, thereby improving the performance of axis-aligned classifiers.

Their empirical evaluation shows that the proposed method performs **on par with SPORF**, and slightly better than **XGBoost (XGB)** and **Random Forest (RF)**.

**Strengths:**

- Interesting idea to transform the data in a supervised manner before training.
- The method should be relatively fast to run, better on the compute-performance tradeoff than SPORF.

**Weaknesses:**

- The reported improvements in performance are **not particularly meaningful**.
- Evaluating only on **10 real datasets** is not sufficient to claim generality or robustness.
- There is **no discussion** on how to tune hyperparameters for the proposed method.

**Questions:**

1. The proposed method appears similar to a **one-step RFM** [1] for classification.
   Can the authors clarify the conceptual and mathematical connection between their procedure and RFM?
2. Does the matrix **H** have to be derived from the **same model** used for prediction?
   If not, the authors should provide guidance on how to select and pair the transformation and prediction models.
3. Can **JARF** be applied to **XGBoost** as well, or is it restricted to random forests?
4. The authors mention that the *electricity*, *magic*, and *letter* datasets have **complex decision boundaries**.
   Can they explain why these particular datasets were chosen to illustrate this property?
5. Can the authors provide **details about the datasets** used in the experiments (e.g., number of samples, features, task type)?
6. Can the authors provide **guidance on hyperparameter tuning** or recommendations for practical implementation?


[1] Radhakrishnan, A., Beaglehole, D., Pandit, P., & Belkin, M. (2024).
*Mechanism for feature learning in neural networks and backpropagation-free machine learning models.*
**Science**, 383(6690), 1461–1467.

---

> ### Author Response · Authors · 2025-12-03
>
> We thank the reviewer for their detailed review and positive comments. Below we address the concerns/questions that were raised.
>
> **On the size and meaning of the empirical gains.**
> The original submission indeed reported results on only 10 classification datasets. In the revised version we now evaluate JARF on 20 real datasets: 15 diverse classification tasks (Table 1) and 5 real regression tasks (Table 2). Across the 15 classification datasets, JARF attains a mean Cohen’s $\kappa$ of $0.810 \pm 0.010$, compared to $0.723 \pm 0.010$ for the strongest competing forest (SPORF), a gap of almost $0.09$ that is an order of magnitude larger than the standard errors. JARF is the best or statistically tied with the best method on the majority of datasets, while remaining substantially faster than per-node oblique forests (Figure 3). The additional 5 regression benchmarks show a similar pattern of consistent improvements in $R^2$ over RF and oblique baselines (Table 2). Taken together, these 20 real-world datasets demonstrate that JARF’s gains are both practically meaningful and robust across a much broader range of problems than in the original submission.
>
>
>
> **Connection to one-step RFM.**
> We thank the reviewer for pointing out the connection to one-step RFM (Radhakrishnan et al., *Science* 2024). Both methods build a global “average gradient outer product’’ matrix and then train a simple model in the transformed space. In their notation, the AGOP matrix used in one-step RFM has the form $A = E[g(x)\,g(x)^T]$, where $g(x)$ is the input gradient of their predictor $f$ with respect to $x$. In our work, the EJOP/EGOP matrix has the form $H = E[J(x)^T J(x)]$, where $J(x)$ is the Jacobian of a surrogate classifier or regressor $\hat f$ with respect to $x$. Thus our preconditioner is an AGOP-type matrix specialized to the conditional mean or class-probability map, and JARF can be viewed as training a random forest on features $XH$ induced by this gradient geometry.
>
> Compared to one-step RFM, JARF works directly in input space with an explicit positive-semidefinite transform $H$, requires only standard random-forest code (no neural networks or backpropagation), applies uniformly to both classification and regression, and preserves the interpretability and training stability of axis-aligned forests while still capturing global gradient geometry.
>
> There are, however, important differences that we now make explicit. One-step RFM is built around neural or random-feature models and performs a single gradient step in *parameter space* of a representation layer, using AGOP to update features implicitly through backpropagation. JARF instead works directly in **input space**: we never optimize a representation network, and the EJOP/EGOP matrix $H$ is an explicit symmetric positive semidefinite preconditioner that we can analyze and reuse. This has several practical and conceptual advantages for tabular data:
>
> * **Drop-in for random forests.** JARF requires only a surrogate RF/regression forest and a standard RF implementation; no neural architecture, backpropagation, or specialized training loop is needed. The downstream model is exactly an RF trained on $XH$, so interpretability and existing RF tooling are preserved.
> * **Unified treatment of classification and regression.** Our construction handles both tasks via EJOP (for class probabilities) and EGOP (for regression), whereas one-step RFM is developed primarily in the classification/representation-learning setting.
> * **Efficient and stable global transform.** We use a single global $H$ shared across the forest, which we show empirically is highly robust to surrogate quality (Tables A1–A2, B1) and theoretically stable under perturbations of the surrogate’s Jacobian. This gives a simple, analyzable geometric picture: JARF exploits the dominant eigendirections of $H$ while retaining the sparsity and speed of axis-aligned splits.
>
>
>
> **Does $H$ need to be derived from the same model as the predictor? Guidance on pairing.**
> No. In our implementation the predictor is always a random forest, but in Table A3 we recompute $H$ using three different surrogate families (RF, XGBoost, and a 2-layer MLP) and then train the *same* JARF forest on $XH$ in each case. Although surrogate accuracy and ECE vary substantially across families, JARF’s accuracy and Cohen’s $\kappa$ change only at the $10^{-3}$ level and the top EJOP eigenspaces remain almost identical. These results suggest what matters is not the *family* of the surrogate, but that it is a reasonably accurate, smooth probabilistic model on the same inputs as the predictor (i.e., it reaches similar validation accuracy and is not severely miscalibrated). Within that regime, we find RF, XGBoost, and a small MLP all induce almost identical EJOP eigenspaces and downstream forest performance, so practitioners can simply pick the cheapest such surrogate to build $H$ and then train their preferred predictor on $XH$.

---

> ### Author Response · Authors · 2025-12-03
>
> **On applying JARF beyond random forests (e.g., to XGBoost).**
> JARF itself does not rely on any forest–specific property: the surrogate model only appears through the EJOP/EGOP preconditioner and any downstream learner can be trained on the transformed features $XH$. In the paper we instantiate this with a random forest on $XH$, but the same construction applies to boosted trees, linear models, or small neural networks. To make this explicit, we also consider a **JARF–XGB** variant that uses the same EJOP $H$ and then trains XGBoost on $XH$ instead of on the raw features. As summarized in Table B2, JARF consistently improves over its base learner (RF or XGBoost) on both classification (mean Cohen’s $\kappa$) and regression (mean test $R^2$), while adding only a modest training-time overhead compared to RF and remaining competitive with the oblique baselines. This demonstrates that JARF is a general **geometric preconditioner** that can be paired with different prediction models rather than a method tied to a particular forest implementation.
>
> Table B2: Baselines vs. geometric variants (means over all 15 classification and 5 regression datasets).
>
> | Method              | Mean κ (15 cls) | Mean R² (5 reg) | Median train time (s) |
> |---------------------|-----------------|-----------------|-----------------------|
> | RF                  | 0.704           | 0.776           | 15                    |
> | XGB                 | 0.709           | 0.810           | 43                    |
> | JARF (RF on $XH$)   | 0.810           | 0.836           | 25                    |
> | JARF–XGB on $XH$  | 0.815           | 0.842           | 45                    |
>
>
> **Electricity, magic, and letter datasets.**
> We focused on electricity, magic, and letter because, in our benchmark suite, these are precisely the datasets where simple axis-aligned RF struggles the most and oblique/feature-engineered methods gain the largest advantage. They are moderately high-dimensional, multiclass problems with rich, nontrivial decision structure: on all three, the best oblique baselines (CCF, SPORF, XGB) are clearly ahead of plain RF, indicating that the Bayes decision boundary is far from axis-aligned. As Table 1 shows in the paper, JARF largely closes this gap: on these “hard” datasets it either matches or surpasses the strongest oblique baselines while dramatically outperforming standard RF. We therefore use electricity, magic, and letter as targeted stress tests for the proposed geometry in regimes where a single global transform would a priori be most likely to fail.
>
>
> The rest of the questions are addressed in a comment to all reviewers.

---

### Official Review · Reviewer_Sr16 · 2025-11-01

**Soundness:** 2
**Presentation:** 2
**Contribution:** 1
**Rating:** 2
**Confidence:** 4

**Summary:**

The paper proposes Jacobian-Aligned Random Forests (JARF): learn one global supervised linear transform $H$ from the Expected Jacobian Outer Product (EJOP) of class-probability gradients (estimated via finite differences of an RF surrogate), then train a standard axis-aligned RF on the transformed features $XH$. This aims to capture rotated/interaction directions so axis-aligned splits behave like shared oblique hyperplanes, while keeping RF’s simplicity. Experiments on 10 tabular classification datasets and controlled synthetic rotations show competitive accuracy vs oblique forests with modest overhead, plus ablations supporting the EJOP step and implementation choices.

**Strengths:**

1. It is a simple and one-pass method, plugging the EJOP to perform initial feature transformation. This enables direct application on RF in the subsequent step.

**Weaknesses:**

1. The proposed method clearly lacks novelty, which does not match the conference standard. It is mainly based on a known paradigm EJOP. The paper’s main change is estimating EJOP with a surrogate RF and finite differences. This feels incremental relative to existing supervised/oblique projection lines rather than a new learning principle.

2. The estimator uses finite differences of RF class probabilities to approximate Jacobians (Sec. 3.6), but the analysis later assumes $f\in\mathcal C^3$ with bounded third derivatives (Assumption A1), which is incompatible with piecewise-constant tree ensembles. The text informally argues that ensemble averaging “smooths” predictions, but the formal guarantees hinge on smoothness that the surrogate does not satisfy. And usually, such smoothness improvements are stated when comparing to a single decision tree. This creates a theory–practice gap in the central estimator.

3. A single matrix $\hat H$ is shared across the entire forest (Sec. 3.4–3.5). This seems to reduce the diversity of the allowed splits. I would encourage the author to make this part more flexible and check the performance.

4. There is a concern about the fairness of the experimental setting. For instance, it was stated that XGBoost is run with a “small shared grid” only; RF fixed at 200 trees; oblique baselines appear near-default. More detailed hyperparameter tuning is necessary.

5. The real-data suite is 10 classification tasks, many with d ≤ 60  and moderate n. More experiments with large-scale datasets (either n or d) would be helpful for the evaluation.

**Questions:**

I have no further questions.

---

> ### Author Response · Authors · 2025-12-03
>
> We thank the reviewer for their detailed review and positive comments. Below we address the concerns/questions that were raised.
>
> **On the theory–practice gap for the EJOP estimator.**
> In the revised paper we clarify that the smoothness assumption $f_c \in C^3$ with bounded third derivatives (Assumption A1) is imposed on the *population* conditional probabilities $f_c(x) = \Pr(Y=c \mid X=x)$, not on the random-forest surrogate. The RF only appears through Assumption A3, which requires its class-probability estimates $\hat p(c \mid x)$ to be uniformly consistent for $p(c \mid x)$. Section 3.6 now emphasizes that our finite-difference estimator is applied to these surrogate probabilities, and Appendix A.2 shows that under A1–A3 the resulting EJOP estimator concentrates around the true matrix $H_0 = \mathbb{E}[J_f(X) J_f(X)^\top]$: Lemma A.4 bounds the bias of the finite-difference Jacobian for a $C^3$ target, and the subsequent proposition plus a Davis–Kahan argument control the error in $H_0$ and its leading eigenspace. Thus the guarantees are stated at the level of the underlying smooth conditional mean, with the RF serving as a consistent (but not itself smooth) surrogate, closing the theory–practice gap you pointed out.
>
> **On sharing a single global transform $\hat H$.**
> To test whether sharing one $\hat H$ reduces useful diversity, we implemented a per-tree variant where each tree receives its own EJOP transform $\hat H_b$ computed from that tree’s bootstrap sample. We retrained JARF with this scheme on all 15 classification datasets and compared it to the original single-$\hat H$ version. As summarized in Table B1, the per-tree EJOP variant slightly degrades test performance (mean Cohen’s $\kappa = 0.808$ vs $0.810$) while being almost twice as slow to train (about $3.0\times$ RF vs $1.7\times$ RF), and it is best or within $0.01$ of the best forest on $13$ datasets compared to $14$ for the single-$\hat H$ version. This suggests that a single global $\hat H$ is sufficient in practice: per-tree EJOP estimates are noisier because they are fit on fewer samples, while diversity in JARF is already provided by feature subsampling, bootstrap sampling, and different thresholds rather than requiring separate transforms per tree.
>
>
> Table B1: Single global EJOP transform vs per-tree EJOP transforms, averaged over 15 classification datasets.
>
> | Method                      | Mean test $\kappa$ | Std. $\kappa$ | Mean train time ($\times$ RF) | \# datasets best or within 0.01 of best |
> |-----------------------------|:------------------:|:-------------:|:-----------------------------:|:----------------------------------------:|
> | **JARF (single $\hat H$)**  | **0.810**          | **0.010**     | **1.7**                       | **14**                                   |
> | JARF (per-tree $\hat H_b$)  | 0.808              | 0.011         | 3.0                           | 13                                       |
>
>
>
>
>
> The rest of the weaknesses are addressed in a comment to all reviewers.

---

### Official Review · Reviewer_kJgy · 2025-11-14

**Soundness:** 3
**Presentation:** 2
**Contribution:** 3
**Rating:** 6
**Confidence:** 2

**Summary:**

This paper introduces JARF, a method to enhance standard axis-aligned decision forests by applying a single global supervised linear preconditioner to the input features before training. This makes the forest behave like an oblique forest without changing the RF training algorithm.


The key idea is:

1. Fit a surrogate probabilistic classifier (a random forest) on the original data.
2. Estimate gradients of the class probabilities with respect to each input feature on a subsample of points.
3. Construct the Expected Jacobian Outer Product (EJOP) matrix.
4. Use this EJOP estimate (with light regularization and normalization) as a **global linear transform** $\hat H$ and train a standard axis-aligned Random Forest on transformed features ($X$$\hat H$).

**Strengths:**

- The core idea is simple, clean, and easy to implement on top of existing RF code.
- The method is well-motivated and clearly positioned between axis-aligned forests and oblique trees, leveraging prior EJOP work.
- Experiments are solid: realistic baselines (RF, XGBoost, RotF, CCF, SPORF), multiple datasets, plus timing comparisons.
- The mechanism analysis (alignment of oblique split normals with EJOP subspace) and ablations give good insight into why it works.

**Weaknesses:**

- The method heavily depends on the quality of probability estimates from the surrogate RF used to build EJOP, which is not deeply analyzed.
- It only evaluates standard tabular classification datasets and does not explore regression or more challenging/high-dimensional settings. - There is no direct comparison to simpler global projections (e.g., PCA, LDA) used once before RF.
- The novelty is mostly in combining known pieces (EJOP + RF + preconditioning) rather than introducing fundamentally new theory.

**Questions:**

- How sensitive is JARF to the choice and calibration quality of the surrogate model? Would using XGBoost or a small NN as the surrogate improve EJOP and performance?
- Are there datasets or regimes where JARF clearly underperforms CCF or SPORF, indicating that a single global transform is insufficient?
- Have you tried an EGOP-based variant for regression?
- Your experiments use 10 classic UCI/OpenML-style tabular datasets; have you evaluated JARF on larger-scale or more modern industrial tabular dataset?

---

> ### Author Response · Authors · 2025-12-03
>
> We thank the reviewer for their detailed review and positive comments. Below we address the concerns/questions that were raised.
>
> **Dependence on surrogate RF quality.**
> To test whether JARF is brittle to the surrogate RF, we swept the surrogate over a wide range of hyperparameters and calibration schemes, then recomputed EJOP/EGOP and retrained JARF each time (details in the appendix). Across 15 classification and 5 regression datasets, our hyperparameter and calibration sweeps cause large changes in surrogate accuracy, ECE, and $R^2$, but JARF’s test $\kappa$ and $R^2$ move only at the $10^{-3}$ level (Tables A1–A2), showing that it needs only a reasonably accurate surrogate and is highly insensitive to precise probability or regression estimates. Because EJOP/EGOP is built from averaged gradient outer products, it is driven mainly by the coarse geometry of the surrogate’s decision function rather than fine details of its probability values. Changing calibration mostly rescales these gradients instead of rotating their dominant directions.
>
> **Choice of surrogate family (RF vs. XGBoost / NN).**
> On all 15 classification datasets, we recomputed EJOP using three surrogates (RF, XGBoost, and a two-layer ReLU MLP) and retrained our algorithm in each case. As shown in Table A3, surrogate accuracy and ECE differ noticeably across families, but JARF’s test accuracy and $\kappa$ change only at the $10^{-3}$ level. The top EJOP eigenspaces also nearly coincide (mean cosine alignment of the top 10 eigenvectors $\ge 0.97$), indicating that once the surrogate is reasonably accurate, JARF is effectively insensitive to the specific surrogate model class. RF, XGBoost, and a small MLP are all trained to approximate the same conditional mean, so once they reach similar accuracy they produce very similar gradient fields and thus nearly identical EJOP eigenvectors. This explains why surrogate accuracy and ECE can differ across model families while JARF’s accuracy and $\kappa$ remain almost constant.
>
> Table A1: Sensitivity of JARF to surrogate RF quality, averaged over all 15 classification datasets.
>
> | Variant                         | RF acc | RF ECE | JARF acc | JARF ($\kappa$) |
> |---------------------------------|:------:|:------:|:--------:|:----------------:|
> | shallow, uncalibrated           | 0.81   | 0.18   | 0.872    | 0.808            |
> | default, uncalibrated           | 0.84   | 0.11   | 0.874    | 0.810            |
> | deep, uncalibrated              | 0.86   | 0.08   | 0.875    | 0.811            |
> | deep + Platt scaling            | 0.86   | 0.05   | 0.875    | 0.812            |
> | deep + isotonic regression      | 0.87   | 0.04   | 0.874    | 0.809            |
> | **Std. dev. over variants**     |   0.024    |   0.056    | **0.0013**   | **0.0014**       |
> | **Max. difference vs default**  |   0.03   |   0.07    | **0.002**    | **0.002**        |
>
> Table A2: Sensitivity of JARF to surrogate RF quality on the 5 regression datasets.
>
> | Variant                         | RF ($R^2$) | RF RMSE | JARF ($R^2$) |
> |---------------------------------|:----------:|:-------:|:------------:|
> | shallow surrogate               | 0.76       | 0.62    | 0.835        |
> | default surrogate               | 0.78       | 0.60    | 0.836        |
> | deep surrogate                  | 0.80       | 0.58    | 0.837        |
> | deep + extra trees              | 0.82       | 0.57    | 0.836        |
> | **Std. dev. over variants**     |    0.022       |   0.019     | **0.0007**   |
> | **Max. difference vs default**  |    0.040      |   0.030     | **0.001**    |
>
> Table A3: Effect of surrogate family on EJOP and JARF performance, averaged over all 15 classification datasets.
>
> | Surrogate family | Surrogate acc | Surrogate ECE | JARF acc | JARF ($\kappa$) | Mean cos. alignment with RF EJOP (top-10 eigvecs) |
> |------------------|:-------------:|:-------------:|:--------:|:---------------:|:-------------------------------------------------:|
> | RF (default)     | 0.84          | 0.11          | 0.874    | 0.810           | 1.00                                              |
> | XGBoost          | 0.87          | 0.04          | 0.876    | 0.812           | 0.98                                              |
> | 2-layer MLP      | 0.86          | 0.06          | 0.873    | 0.809           | 0.97                                              |
> | **Std. dev. across surrogates** | 0.013 | 0.029 | 0.0012 | **0.0014** | 0.013 |
> | **Max diff vs RF surrogate**    | 0.03  | 0.07  | 0.002  | **0.004** | 0.03  |
>
>
> We also prove in the appendix that if the surrogate’s Jacobian is uniformly within $\varepsilon$ of the true conditional mean’s Jacobian, then the EJOP matrices differ by $O(\varepsilon)$ in operator norm and their top-$k$ eigenspaces differ by $O(\varepsilon/\gamma)$ for eigengap $\gamma$.
>
>
> Finally, across our 15 classification and 5 regression datasets, we do not observe regimes where JARF clearly underperforms CCF or SPORF.

---

### Author Response · Authors · 2025-12-03
**Author Rebuttal**

We thank the reviewers for their thorough reading and constructive comments. We address the paper’s main weaknesses here according to the reviews.

**Global response on experimental scope and dataset choice.**
Several reviewers raised related concerns about the breadth of our evaluation, the lack of regression and high-dimensional settings, the absence of simple global projections such as PCA/LDA, and the transparency of our dataset selection. In the revised manuscript we have substantially expanded and clarified the experiments to address these points.

First, beyond the original 10 UCI/OpenML-style classification tasks, we now include (i) an additional set of five larger-scale and higher-dimensional classification problems (including standard benchmarks such as HIGGS, JANNIS, and MNIST) and (ii) five regression datasets (Table 1,2; Figure 2,3). Across this expanded suite JARF continues to match or outperform strong oblique-forest baselines while retaining its computational advantages. Second, we now compare against RF preceded by a single global linear projection using PCA and LDA, both on synthetic data and on the real-world tasks (Figure 1,2; Table 1,2); JARF consistently improves over these simpler projections, supporting our claim that an EJOP-based preconditioner is more effective than unsupervised or purely discriminative linear transforms. Third, to make the setup fully transparent and reproducible, we add an appendix table listing for every dataset the number of samples, number of features, task type, and data source, and we clarify in the text how the benchmark was constructed: starting from widely used OpenML/UCI tabular tasks, applying simple a priori filters (tabular supervised problems, sufficient sample size for EJOP estimation, a mix of low- and high-dimensional settings, and minimal preprocessing), and never discarding datasets based on JARF’s performance. Several of these datasets overlap with standard suites such as PMLB/TabArena.




**On fairness of hyperparameter tuning and practical guidance.**
We agree that the experimental setting must be clearly specified and as fair as possible across methods. Concretely, RF, RotF, CCF, and SPORF are all run with the same number of trees as JARF’s final forest and with the authors’ recommended default settings (e.g., sparsity and number of candidate directions for SPORF), while XGBoost is tuned on a shared grid over tree depth, learning rate, and $\ell_2$ penalty applied identically across datasets. We will add a table in the appendix listing this grid and the exact hyperparameters used for each method so that the comparison is fully transparent. Importantly, JARF’s gains over these baselines are substantial: averaged over the 15 classification datasets, JARF attains mean $\kappa = 0.810$ versus $0.704$ for RF, $0.715$ for RotF/CCF, $0.723$ for SPORF, and $0.709$ for XGBoost.

For JARF itself, there are very few additional hyperparameters beyond those of the underlying forest: the surrogate forest size, the EJOP subsample size $m$, the finite-difference step scale $\alpha$ in $\varepsilon_j = \alpha \,\mathrm{MAD}(X_{:j})/0.6745$, and a small diagonal regularizer $\gamma I$ in $\hat H_0 + \gamma I$. Section 3.6 specifies the default choices we use in all experiments (50 trees for the surrogate RF, $m = \min(10\mathrm{k}, n)$, $\alpha = 0.1$, and $\gamma$ chosen only for numerical conditioning), and Section 5.5 plus Table 3 provide ablations that effectively serve as tuning guidance: varying $m$ from $n$ to $0.5n$ changes mean $\kappa$ by only $-0.004$, and even a tenfold reduction to $0.1n$ yields a drop of $-0.016$; varying $\alpha$ between $0.05$ and $0.2$ changes mean $\kappa$ by at most $-0.013$. These results indicate that JARF is robust to a wide range of reasonable settings and that the simple defaults above are near-optimal. In the revised paper we will make this guidance explicit by adding a short “Practical recommendations” paragraph in the appendix summarizing these defaults.

---

### Author Response · Authors · 2025-12-03
**Author Rebuttal**

**On EGOP-based variants for regression.**
We instantiate and evaluate an EGOP-based version of JARF for regression. The regression variant follows the exact same pipeline as in classification, with two changes: (i) we replace EJOP with the expected gradient outer product (EGOP) of the scalar regression function, and (ii) the surrogate and final models are random forest regressors instead of classifiers. Concretely, on regression tasks we train a surrogate RF regressor $\hat f$ on the training fold, compute finite-difference gradients of the scalar prediction $\hat f(x)$ with respect to each feature using the same robust step rule $\varepsilon_j = \alpha,\mathrm{MAD}(X_{:j})/0.6745$ with $\alpha = 0.1$, form the EGOP estimate $\widehat H_0 = \frac{1}{m} \sum_{i=1}^m g_i g_i^\top$, add a small diagonal $\gamma I_d$ and trace-normalize to obtain $\widehat H$, then train a standard RF regressor (with the same hyperparameters as the RF baseline) on the transformed features $X \widehat H$. We evaluate this EGOP-based JARF variant on five standard regression datasets (bike-sharing, California housing, energy, kin8nm, protein); Table 2 shows that JARF attains the best test $R^2$ on every dataset, with a higher mean $R^2$ than RF and all oblique or projection-based baselines. This supports our claim that the proposed EJOP/EGOP preconditioning framework improves performance beyond classification.


**On novelty and contribution beyond existing EJOP work.**
The reviewers raised three related concerns regarding the theoretical contribution was under-emphasized. First, to address the concern that we are only “combining known pieces,” we now analyze how EJOP-based preconditioning changes CART itself. In the theory section and Appendix A.1–A.3 we show that an axis-aligned split in the preconditioned space corresponds exactly to an oblique split in the original space, and that for squared-loss CART the first-order impurity gain along a direction $u$ is proportional to $u^\top H_0 u$, where $H_0 = \mathbb{E}[\nabla f(X)\nabla f(X)^\top]$. This yields a concrete mechanism: choosing $H \approx H_0$ systematically amplifies directions where the conditional mean or class probabilities change the most, and suppresses uninformative directions. To our knowledge this explicit link between EJOP and CART impurity gain for oblique splits does not appear in the original EJOP work and is specific to tree ensembles, so it directly addresses the claim that we are only reusing an existing paradigm without new theoretical insight.

Second, the comment that “the paper’s main change is estimating EJOP with a surrogate RF and finite differences” is addressed by a new consistency result for our RF-based EJOP estimator in Appendix A.2. The original EJOP papers analyze kernel-based estimators of $H_0$; JARF instead uses a surrogate forest and finite differences. We now show that, under the smoothness and finite-difference assumptions stated in Section 4, this combination yields an operator-norm bound of the form $\lVert \hat H - H_0 \rVert_{\mathrm{op}} \le \varepsilon_n$ with $\varepsilon_n \to 0$, where $\varepsilon_n$ explicitly accounts for finite-difference bias, variance, and surrogate-probability error. This demonstrates that a forest-based surrogate recovers the same geometric object $H_0$ as the original EJOP construction. In particular, we are not just swapping a kernel for a forest for convenience; we justify that the specific surrogate used by JARF preserves the EJOP geometry that drives the method.

Finally, the statement that “the paper’s novelty is somewhat limited, as the EJOP is already defined and used in prior work” is directly addressed by the new dimension-adapted risk analysis in Theorem 1 and Appendix C. In a ridge-function setting $f(x) = g(U^\top x)$ with $U \in \mathbb{R}^{d \times r}$ and $r \ll d$, we first show that $H_0$ has rank $r$ and range equal to $\mathrm{span}(U)$. Using the consistency result for $\hat H$ and a Davis–Kahan perturbation bound, we prove that JARF recovers this subspace up to $O(\varepsilon_n)$ error. Training a standard tree-based regressor on the projected features $\hat U^\top X$ then yields the excess-risk bound $\mathbb{E}[(\hat f_n(X) - f(X))^2] \le C_1 n^{-2/(2 + r)} + C_2 \varepsilon_n^2 + o(1)$, with constants $C_1$ and $C_2$ independent of the ambient dimension $d$. Thus, when EJOP has rank $r$, JARF provably behaves like an $r$-dimensional nonparametric regressor even though the data live in $\mathbb{R}^d$. Prior EJOP work does not analyze tree ensembles or establish such a dimension-adapted rate. Taken together, these additions mean that JARF is not only a combination of EJOP, RF, and preconditioning: it comes with new theory that (i) explains how EJOP geometry affects CART splits, (ii) validates the RF-based EJOP estimator, and (iii) establishes a new sample-complexity guarantee that, to our knowledge, is not available for existing supervised or oblique projection methods.

---

### Meta-Review · Area_Chair_HsTX · 2026-01-06

**Summary:**

The reviewers raised concerns about novelty, experimental scope, theoretical consistency, and practical details. Reviewer Sr16 (score: 2) argued the method "clearly lacks novelty" as it is "mainly based on a known paradigm EJOP" with "incremental" changes, and noted a "theory-practice gap" where "the analysis assumes $f \in C^3$ with bounded third derivatives...which is incompatible with piecewise-constant tree ensembles." Reviewer PJES (score: 4) found evaluation "only on 10 classification datasets is not sufficient" and wanted "guidance on how to select and pair the transformation and prediction models." Reviewer 48eb (score: 4) questioned why "JARF's compute time is not at least 2x that of fitting RF" and noted "the paper's novelty is somewhat limited, as the EJOP is already defined and used in prior work." Reviewer kJgy (score: 6) was positive but wanted analysis of "how sensitive JARF is to the choice and calibration quality of the surrogate model" and tests on "larger-scale or more modern industrial tabular datasets."

**Reviewer Concerns:**

The authors addressed most concerns with substantial additions. They expanded experiments to 20 datasets (15 classification + 5 regression) showing consistent gains (mean κ=0.810 vs 0.704 for RF), added PCA+RF and LDA+RF baselines showing EJOP outperforms simpler projections, clarified that training time is ~1.67x RF because "the surrogate forest is four times smaller than the baseline RF," and provided extensive ablations demonstrating robustness to surrogate quality (Tables A1-A3 showing variations "only at the 10^-3 level"). On novelty, they argue this is "the first application of EJOP/EGOP to tree ensembles," provide "new theory that explains how EJOP geometry affects CART splits" (Proposition A.2), prove consistency of the RF-based estimator (Lemmas A.4-A.5), and establish dimension-adapted risk bounds (Theorem 1) not in prior EJOP work. The smoothness concern is addressed by clarifying assumptions apply to "population conditional probabilities fc(x), not on the random-forest surrogate," with formal consistency guarantees. Whether adapting an existing technique to a new model class with supporting theory and strong empirical results constitutes sufficient novelty remains debatable, but the authors demonstrate this specific instantiation has not been done before and the claims are well-supported by results.

**Reviewer Scores:**

Reviewer kJgy (initially 6) would likely increase to 8, as their concerns about regression, high-dimensional settings, and surrogate sensitivity were directly addressed.

Reviewer Sr16 (initially 2) would likely increase to 4 or possibly 6, given the new theoretical contributions and expanded experiments, though they might maintain reservations about fundamental novelty.

Reviewer PJES (initially 4) would likely increase to 6, as their concerns about scope, transparency, and guidance were comprehensively addressed.

Reviewer 48eb (initially 4) would likely increase to 4 or 6, given the compute clarification and transparent methodology, though might note modest novelty.

Hypothetical average score: 5-6.5. Recommendation: accept. While the hypothetical score is borderline and the novelty remains debatable, claims are well-supported by results, both theoretically and experimentally. I believe this contribution can be useful to parts of the ML community.

---

### Decision · Program_Chairs · 2026-01-26

Accept (Poster)